# Challenges and limits of mechanical stability in 3D direct laser writing

Elaheh Sedghamiz[1], Modan Liu[1] & Wolfgang Wenzel [1✉]

Direct laser writing is an effective technique for fabrication of complex 3D polymer networks using ultrashort laser pulses. Practically, it remains a challenge to design and fabricate high performance materials with different functions that possess a combination of high strength, substantial ductility, and tailored functionality, in particular for small feature sizes. To date, it is difficult to obtain a time-resolved microscopic picture of the printing process in operando. To close this gap, we herewith present a molecular dynamics simulation approach to model direct laser writing and investigate the effect of writing condition and aspect ratio on the mechanical properties of the printed polymer network. We show that writing conditions provide a possibility to tune the mechanical properties and an optimum writing condition can be applied to fabricate structures with improved mechanical properties. We reveal that beyond the writing parameters, aspect ratio plays an important role to tune the stiffness of the printed structures.

[1] Institute of Nanotechnology (INT), Karlsruhe Institute of Technology (KIT), Hermann-von-Helmholtz-Platz 1, 76344 Eggenstein-Leopoldshafen, Karlsruhe, Germany. ✉email: wolfgang.wenzel@kit.edu

Three-dimensional direct laser writing (3D-DLW) is an indispensable tool for high-accuracy structuring and fabricating arbitrary 3D micro- and nano-objects via a nonlinear absorption induced polymerization process[1–3]. Over the past decade, this technology has become a well-established lithography tool for fabricating a wide variety of 3D structures directly on functional substrates[4,5]. This technique enabled a wide range of applications in photonics[6], microfluidics[7], as well as the generation of mechanical microstructures[8], and cell scaffolds[9]. Despite recent advances in manufacturing techniques and their applications, different limitations and challenges remain. For example, new photoresist formulations for fabricating conducting polymer devices[10,11], integrating new biocompatible materials[12], the integration of multiple materials in the same structure[13], increasing the structuring rate[14,15], and obtaining sufficient and controllable stiffness of the fabricated structure[16,17]. Moreover, the relatively large feature size and limited resolution of DLW compared to electron-beam lithography (EBL) and focused ion beam milling have limited its application to deep sub-micron lithography. Several strategies have been applied in order to decrease feature size driven by an interest in shrinking devices to nanometer scales, such as, introducing mobile quenching molecules[18] and implementing a second annular inhibiting laser, i.e., stimulated emission depletion (STED) lithography[19]. To the best of our knowledge, the smallest feature size and the separation achieved to date are 7 and 33 nm, respectively[20].

Recently, there has been a growing effort to bring direct laser fabrication from a laboratory curiosity to a versatile tool for the fabrication of various micro- nano-structures of 3D materials for functional devices[21–23]. It has been observed that, while fabricating structures with higher stiffness is favorable in many applications[11], in some applications, such as tissue engineering, 3D scaffolds with smaller Young's modulus result in a better functionality as compared to stiffer structures[24]. In addition, structural modifications induced by shrinkage or insufficient stability during drying must be considered in the development step[25,26]. Overall, the morphology and stiffness of the fabricated micro-/nano-structure play a key role in the functionality of the structure[13,16] and it is important to understand how both parameters can be controlled in the context of the DLW process parameters. This is of particular relevance for improved manufacturing of delicate structures. Experimental studies in this realm encounter serious challenges, as it is not easy to control all parameters that influence the DLW process[27] or to observe this process on the nanoscale in operando. In addition, difficulties in reliable fabrication of nanoscale specimens in combination with strain measurement complications make mechanical experiments challenging[28].

To complement the experiment, computer simulations offer a time-resolved virtual analog of the 3D writing process, in particular at small scales, and promise insights into the details of this process that are difficult to obtain by experiment. Given its favorable cost, simulations allow for systematic studies of the variation of structure and stiffness of the fabricated polymer network[29,30] as a function of the process and materials parameters, which—in the long-term—potentially enables virtual design/screening of materials and process conditions. There have been several studies attempting to emulate the two-photon polymerization conditions and extract the role of various involved process and material-dependent parameters[31–33]. Analytical models have been developed that model termination via radical combination and radical trapping to investigate the effect of photoinitiator concentration, light intensity, and oxygen quenching on the polymerization process and time-dependent conversion[31–33]. While these numerical models are able to capture some aspects of laser-induced polymerization, their limitation is the lack of molecular-scale information and parameters that must be fitted to experiment. This limits the long-term potential of these methods and their predictive value.

With the continuous improvement of computational resources, molecular dynamics (MD) simulation, which represents both the molecular structure of the system and the dynamics of the process, has emerged as an increasingly powerful tool to model complex structure formation processes. This approach is capable of modeling many process parameters and yields structural models that can be analyzed with respect to their mechanical properties[34–37] but remains limited with respect to system size and the time scales that can be treated[30,38,39]. To date, there have been few studies that systematically investigated the influence of the initial composition and process parameters on the formation of the polymer network and its properties. Wang et al.[40] have used this approach in order to model continuous liquid interface production (CLIP) 3D printing technique aiming to investigate the effects of elastic, capillary, and friction forces on the quality of the shape of the 3D printed objects. Karnes et al.[41] studied the cross-linking of acrylate polymer networks using a reactive MD simulation approach with an all-atom force field. Their study links the monomer conversion efficiency with macroscopic properties, e.g., gel point conversion, illustrating the ability of molecular mechanics as a useful tool in the rational design of photo-polymerized resins.

Nevertheless, the accurate correlation between the writing condition, architecture, and mechanical properties of the fabricated polymer networks in DLW remains elusive and no theoretical model has been developed to dispel this issue. In this work, we develop and apply a molecular dynamics (MD) based protocol to simulate direct laser writing of 3D polymer networks starting from acrylate-based monomers, where monomers stick irreversibly with respect to experimental reaction rate constants by focusing on an experimentally important system, tri-acrylate family of photoresist, described in the method section. We employ a coarse-grained MD simulation approach to explore the mechanical stability of 3D printed polymer networks of different aspect ratios applying different laser powers and exposure times and to explain how the structural and mechanical properties of the printed polymer network are correlated to DLW condition and their aspect ratio.

## Results

**Characterization of the printed polymer networks.** The degree of monomer conversion i.e., the fraction of bound functional groups as compared to the overall number of functional groups is an important parameter for polymerization reactions, especially in the DLW process. In polymer chemistry, it has been found that there is a saturation for monomer conversion as a function of the reaction time[41]. In the case of DLW, different authors reported different degrees of conversion for commercial and noncommercial photoresists[42,43]. While the control of the degree of polymerization in the voxel during the actual writing process is of utmost importance for further improvements of DLW, it has been a challenge to directly measure the level of monomer conversion in the printed area. A few studies investigate the effect of the writing condition on the properties of the printed polymer network[28,43–47]. By using different detection instrumentations, for noncommercial resists based on a mixture of two different tri-acrylates, a monomer conversion degree of 60–75% has been obtained while for organic-inorganic hybrid resists, the degree of conversion has been reported to be 35–75%[43,44]. Nevertheless, it is clear that the solidification and formation of chemically stable networks occur at relatively low conversions well below 100%.

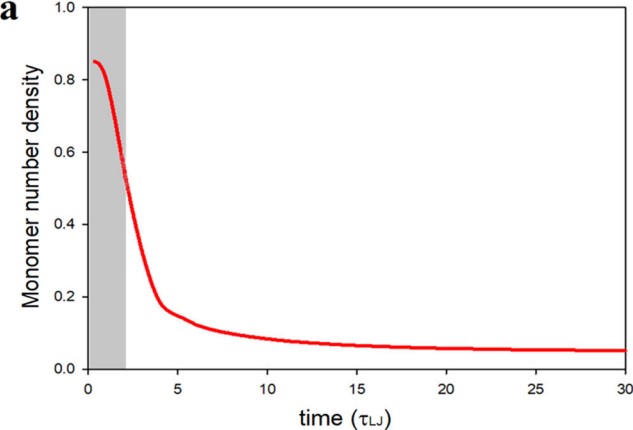

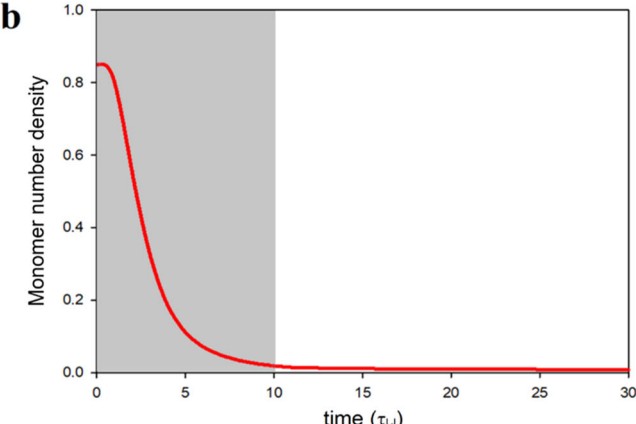

**Fig. 1 Kinetics of monomer consumption. a** Exposure times $= 2\,\tau_{LJ}$, **b** exposure times $= 10\,\tau_{LJ}$. The gray area shows exposure time.

We first consider different scenarios for the polymerization time vs. the exposure time, the former being defined as the time where a substantial polymerization can be observed, even after no new free radicals are generated anymore. To illustrate this effect, Fig. 1 shows the kinetics of monomer consumption in one voxel when the exposure time is 2 and $10\,\tau_{LJ}$. For the short exposure time, the polymerization reaction occurs from 0 to $10\,\tau_{LJ}$ while for the longer exposure time it occurs between 0 and $15\,\tau_{LJ}$. For the simulation with an exposure time of $2\,\tau_{LJ}$, a significant amount of polymerization occurs after the exposure (dark polymerization) which is the common scenario in DLW of all tri-acrylate-based photoresists[33]. In this scenario, some free monomers are left within the exposure area while for long exposure times; all monomers are being consumed during the exposure time. We note that the dynamic of the DLW is accelerated in our simulations because of the nature of the employed polymer model and converting the LJ time unit to the actual unit results in unrealistic exposure times and therefore we report the simulation data only in LJ units throughout this manuscript.

To systematically study the degree of conversion (DC) as a function of laser power and exposure time, cubic polymer networks of size $40\sigma \times 40\sigma \times 40\sigma$ with an initial volume fraction of 85% monomers were fabricated by uniform irradiation and applying periodic boundary condition in all directions. As expected, the monomer conversion increases with higher laser power or, correspondingly, higher exposure times and saturates towards high writing powers as shown in Fig. 2a. We find that the laser power has a stronger influence on the degree of monomer conversion than the exposure time. For instance, increasing the laser power from 14 to 45 mW increases the degree of conversion

from 49 to 74%, while at the subsequent increments to 64 mW, the degree of monomer conversion only increases by ~1 percentage points. Hence, it is sufficient to apply moderate laser powers of 40–45 mW to reach a high monomer conversion ratio.

In addition, Fig. 2a shows that the exposure time has a more pronounced effect on monomer conversion at lower laser powers. Figure 2b compares the %DC obtained from the simulation with the experimental data for tri-acrylate-based photoresist[43,44]. The %DC values obtained from the simulation are 5–15% higher than the experimental values. Agreement with the experiment is particularly good at very low and very high laser powers, where simulation results in less than 3% over-prediction in the degree of monomer conversion. Furthermore, the overall trend shows a very good coincidence with the experiment trend. Experimentally, it has been observed that the progression of the double-bond conversion at increasing laser powers confirms the second-order dependency that is also predicted by our model. It is worth noting that the degree of monomer conversion reported experimentally depends also on the applied measuring technique. Overall, the presented model is able to reproduce the experimental results of %DC qualitatively and quantitatively within an acceptable deviation range.

Having generated a number of protocols for different exposure conditions, we are now able to characterize the mechanical properties of the polymerized network. It has been found that the laser power applied for photo-polymerization critically affects the mechanical properties of the final structure[17]. To study the mechanical properties, we fabricate cubic polymer networks applying different laser powers and constant writing velocities. We find that the increase in the laser power leads to a higher cross-linking density as shown in Fig. 3a. The degree of monomer conversion is increasing from 49 to 75% for the applied laser powers.

The mechanical properties of the printed networks are studied by simulating stress–strain dependencies and calculating Young's modulus (E) from the slope of the stress–strain curve under different writing conditions, as discussed in "Methods". As shown in Fig. 3b, stress increases by increasing the laser power which is consistent with the behavior of the tri-acrylate photoresist under similar conditions[17,28]. Stress–strain curves do not illustrate a linear elastic behavior but they show a degree of hyperelasticity that is the typical behavior of rubbery materials under the mechanical test reflecting a two-region behavior (linear and nonlinear)[48]. The first region, between 0 and 0.2, with the smaller slope, corresponds to the increase of the stress due to the recoiling of the polymer chains and the part with the larger slope corresponds to the FENE bond stretching[17,49]. The mechanical properties of the printed networks can be better quantified by estimating E from the slope of the stress–strain curve.

To compare the results of the simulation with the experimentally measured Young's modulus for the tri-acrylate photoresists, experimental data points are shown with the green dots in Fig. 3c. Experimentally, it has been reported that while the progression of the double-bond conversion at increasing laser energies confirms the second-order dependency as we have also observed in our simulations (Fig. 2), increasing the laser power leads to an approximately linear increase in Young's modulus in the laser range from 9 to 25 mW[50]. Lemma et al.[17] also performed a linear fit of E values obtained for the laser powers below 20 mW. We observed that while for laser powers below 25 mW, E increases linearly as a function of the laser power; it reaches a threshold for higher laser powers. Therefore, the progression of Young's modulus shows a saturation behavior as the double-bond conversion did. As expected, there is a direct relationship between the degree of monomer conversion and Young's modulus of the fabricated structure. Such behavior has not been

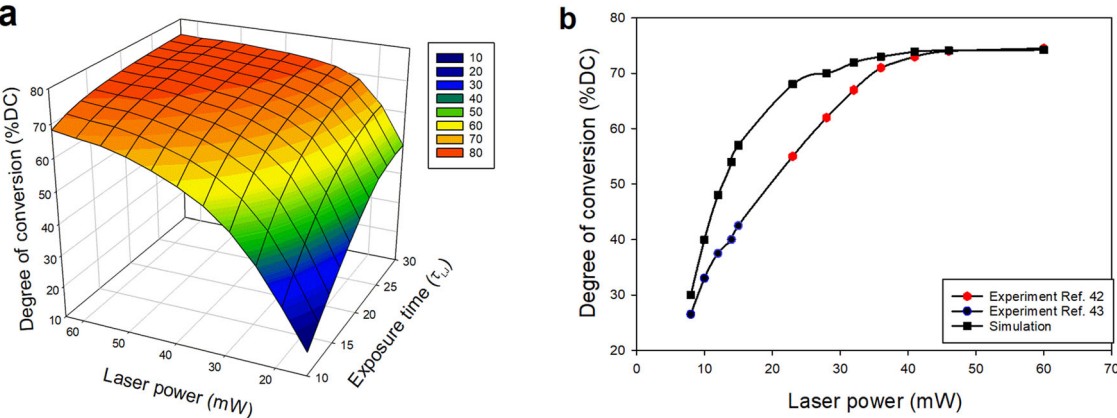

**Fig. 2 Degree of monomer conversion. a** Degree of monomer conversion with respect to exposure time and laser power. **b** The progression of degree of monomer conversion by increasing the laser power derived from simulation and experimental measurements.

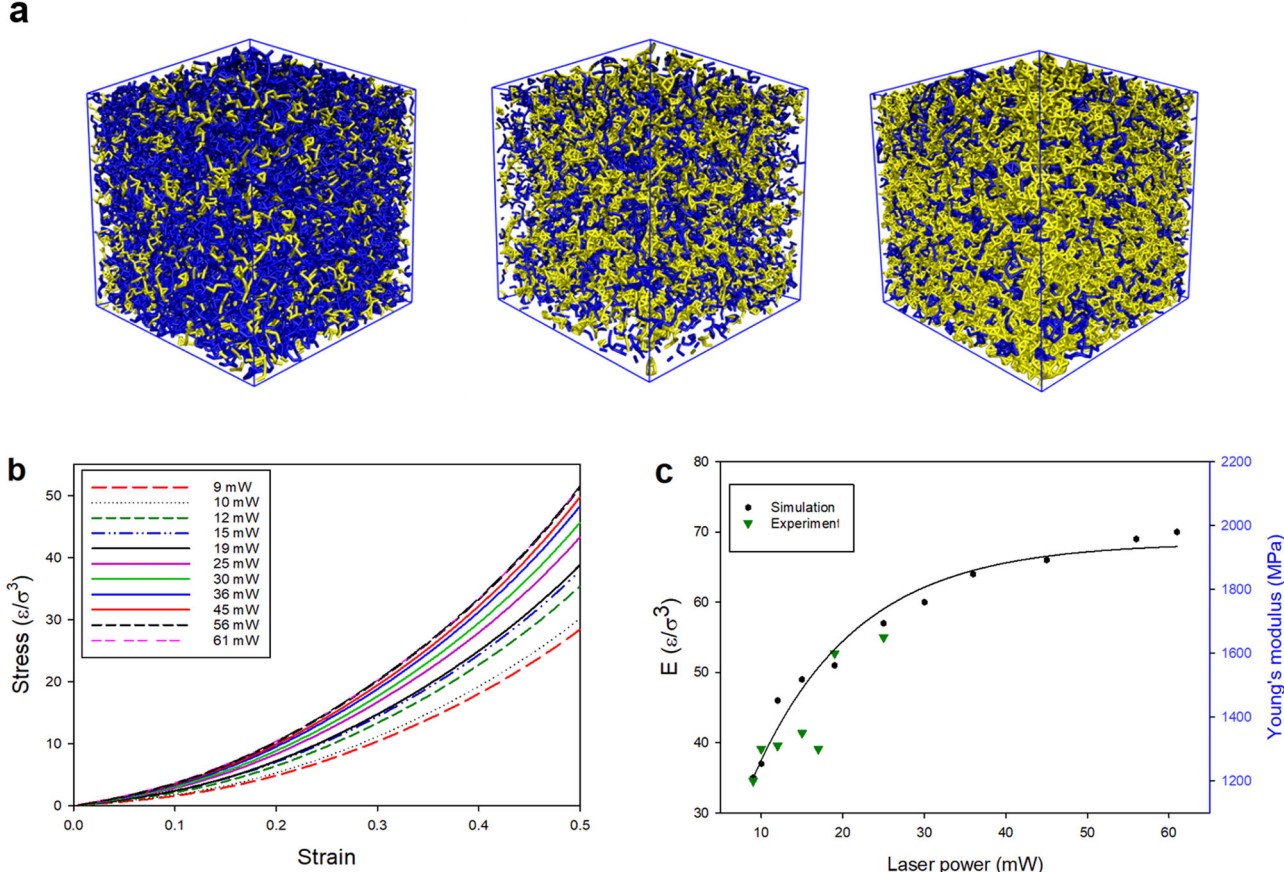

**Fig. 3 Effect of laser power. a** 3D polymer blocks printed applying different laser powers, (yellow regions represent cross-linked monomers) illustrate the increase in the cross-linking density as the laser power increases from 10 to 46 mW. **b** Stress–strain curve for the printed networks with different laser powers. **c** Young's modulus (E) as a function of the laser power from simulation and experimental measurements (one experimental data set from ref. [50] has been selected because of the similarity of the employed photoresist to our model photoresist).

observed experimentally since experimental measurements of Young's modulus were only possible for structures fabricated within a limited range of laser power (9–25 mW) due to technical problems in measuring the mechanical properties[50]. This observation illustrates the benefit of simulations that makes it possible to study a broader range of conditions in the DLW process. It is worth noting that in very high laser powers (higher than 65 mW) it is impossible to obtain experimental data as

overheating causes damage to the polymerized structure[42]. Figure 3c also suggests that a polymer network with near-maximal stiffness can be obtained by applying moderate laser powers between 35 and 40 mW. Our simulation shows that applying such laser power results in a polymer network with ~70% degree of monomer conversion (degree of monomer conversion is increasing from 41% for the lowest laser power to 75% for the highest one).

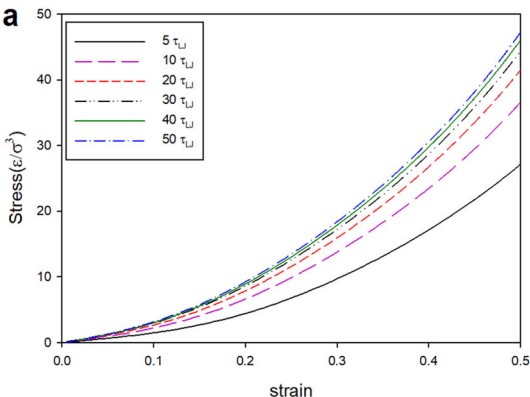 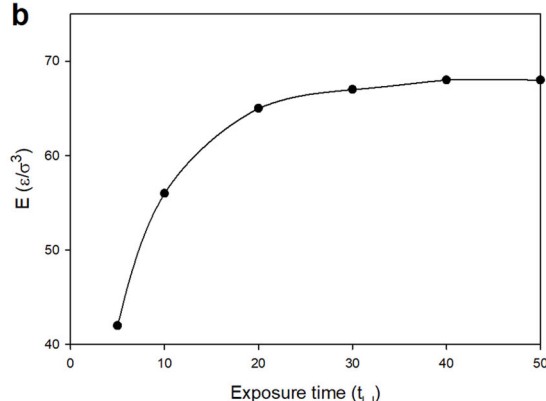

**Fig. 4 Effect of exposure time. a** Stress–strain curve for printed blocks applying different exposure times. **b** Young's modulus (E) calculated from the slope of the stress–strain curves for different exposure times.

**Effect of writing velocity on the properties of printed polymer networks**. Mechanical properties of printed objects need tuning to enhance the performance in some applications. For instance, in tissue engineering, Young's modulus of the synthesized scaffold must vary for different kinds of cells to regenerate better[16,51–53]. One direct approach for modifying the modulus is to change the writing laser power, as we illustrated earlier. However, in the experiment, sometimes it is easier to change the writing velocity than the laser power during the fabrication process of a single object[54]. To investigate the effect of exposure time on the mechanical properties of the printed objects, we fabricate cubic polymer networks by applying different exposure times (different writing velocities) and constant laser power (46 mW). Therefore, the decrease in the exposure time leads to a decrease in the number of active monomers and lower cross-linking density. Then we deform the fabricated networks to obtain the stress–strain curve and calculate Young's Modulus using the same procedure explained in the previous section. As shown in Fig. 4, Young's modulus increases with the exposure time, which is the direct result of increasing the degree of monomer conversion, which increases from 38% for $t = 5\tau_{LJ}$, to 56% for $t = 10\ \tau_{LJ}$, to 73% for $t = 30\ \tau_{LJ}$ and 74% for $t = 50\ \tau_{LJ}$. Young's modulus increases by about 60% (from ~42 $\epsilon/\sigma^3$ to ~68 $\epsilon/\sigma^3$) by increasing the exposure time from 5 $\tau_{LJ}$ to 50 $\tau_{LJ}$. It reaches a threshold value at 30 $\tau_{LJ}$ which indicates the saturation in cross-linking density. It is due to the fact that all activated monomers polymerize in the early stage of the exposure time and further exposure leads to an increase in cross-linking density, however, in longer exposure times, the high steric effect within the formed 3D network do not allow for further cross-linking between the polymer chains.

**Stiffness of the printed polymer rods**. Recent developments in nanoscale devices take advantage of many complex patterns with high aspect ratio structures in their design[46,55–58]. High strength and high deformability are essential properties for the design and reliability of novel nanodevices[59]. The stress–strain curves, which result from a collective effect from all the particles, do not depend on the aspect ratio of the structure. To investigate the stiffness and deformability of small printed objects, we are more interested in some local events to understand how polymer networks of different aspect ratios behave under an external loading force as these local events can lead to crack formation. Therefore, nano-rods with different aspect ratios (length to diameter ratio) i.e., 10σ, 20σ, 30σ, and 40σ cross-section diameter and height of 100σ were fabricated (see Fig. 5) in a monomer pool of 80σ × 80σ × 240σ applying different laser powers ranging from 10 to 50 mW.

The printed rods were deformed under the influence of an external force (details are provided in "Methods"). The maximum deflection versus the cross-section diameter for different laser powers is shown in Fig. 5e. It shows that the deflection decreases as the cross-section diameter increases for all applied laser powers, indicating that printed objects with higher aspect ratios have lower stiffness and hence higher flexibility (deformability). Nevertheless, it is also essential to investigate the probability of crack formation for each rod in the deformation test.

To understand how the printed samples behave under an external force, we investigate some local properties, i.e., particle strain inside the rods at their maximum deformation. The distribution of atomic strains for all fabricated samples at their maximum deflection is illustrated in Fig. 6. All microstructural analysis and visualization of atomic strains are performed using the open-source visualization tool OVITO[60]. The highly localized, high-strain region is observed for all samples with high aspect ratios (height/width) $\frac{h}{w} = 10$ and 5 which get smaller by increasing the laser power. This region of high strain in high aspect ratio structure is likely to yield the onset of a fracture for all applied laser powers. Note that the actual formation of a fracture cannot be observed in our simulations since the structures are small and our model does not implement bond breaking. For samples with $\frac{h}{w} = 3.3$ large areas of moderate atomic strain and small regions of high strain are observed at low laser powers that disappear by increasing the laser power. For samples for rods with $\frac{h}{w} = 2.5$, a very small high-strain region can be detected, instead there exist low to moderate strain regions spreading throughout the deformed areas. Furthermore, for structures with smaller aspect ratios, no significant difference in the strength and distribution of atomic strain can be seen when the laser power is increased. Therefore, the stiffness of high aspect ratio rods is mainly independent of the applied writing condition i.e., laser power, suggesting the existence of a size threshold for which the expected writing condition sensitivity does not apply. However, for nano-rods with larger aspect ratios i.e., 10, 5, and 3.3, laser power affects the stiffness of the rod by decreasing the high-strain areas and tailoring the rod stiffness from weak to strong, depending on the applied laser power.

We have presented a molecular dynamics simulation method to print 3D objects voxel by voxel from activated monomers and investigate the structural and mechanical properties of the fabricated 3D polymer networks. Based on the mechanism of free-radical polymerization, 3D polymer networks were fabricated from the monomer pool in voxels and the formation of networks is governed by reaction rates constants and defined by classical potential energy functions and types of monomers participating

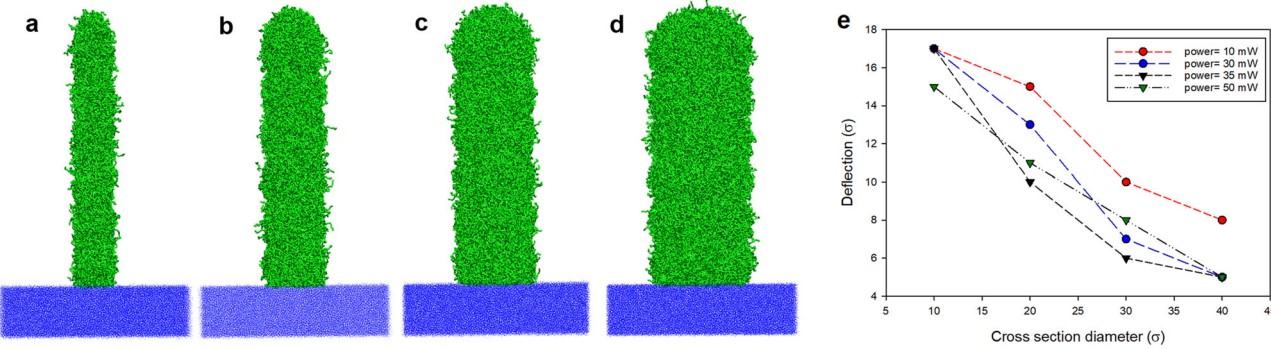

**Fig. 5 Printed rods with different aspect ratios. a** h/w = 100σ/10σ = 10, **b** h/w = 100σ/20σ = 5, **c** h/w = 100σ/30σ = 3.3, and **d** h/w = 100σ/40σ = 2.5. **e** Maximum deflection of printed rods vs. cross-section diameter for different applied laser powers. Dashes are shown only for the help of eye.

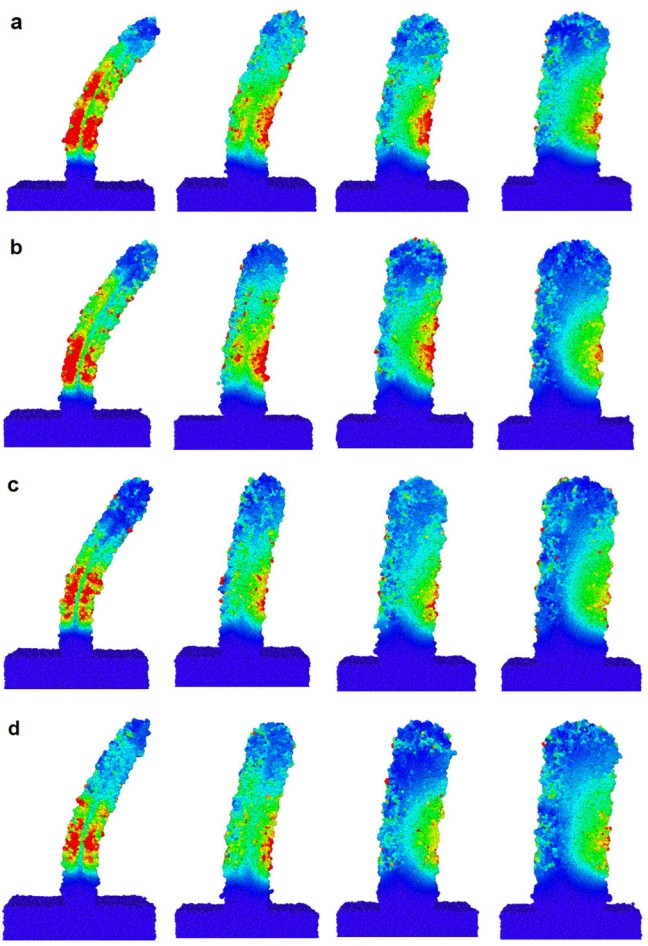

**Fig. 6 Distribution of atomic strain in printed rods of different aspect ratios (shown in Fig. 5) with particles colored according to their strain for laser powers. a** 10 mW, **b** 30 mW, **c** 35 mW, and **d** 50 mW. Color map is for atomic strain and the rods are sliced through the slab centers.

in the elementary reactions. Therefore, the formation of the polymer networks in our method emulates a real polymerization reaction in the DLW process and makes it possible to create networks with topologies reminiscent of realistic networks. It has been shown that the degree of monomer conversion and therefore mechanical properties of the polymerized structure can be controlled by changing the laser power and to a lesser extent by changing the writing velocity as long as the exposure time below the threshold value is applied. Our simulations revealed a second-

order dependency between the Young's modulus and laser power contrary to the linear dependency that has been reported previously based on the experimental measurements, which was due to the technical limitations in experimental measurements.

Overall, it has been found that to fabricate structures with high mechanical properties an optimum writing condition should be applied. While fabricating high aspect ratio structures is desired for many applications, our work highlights the limit in fabricating high aspect ratio structures in DLW, which could not be completely overcome by changing the writing conditions. We showed that the effect of laser power becomes less and less important when the aspect ratio of the printed objects increases.

As experimental methods to observe the 3D printing process on the nanoscale are limited, we believe simulations based on microscopic models can help optimize the printing conditions especially for smaller feature sizes. We note that our approach can be expanded for any photoresist to study the effect of writing conditions and architecture on the mechanical properties of polymer networks and may thereby enable a substantial reduction of experimental characterizations. It can pave the way for more computational studies in this field and is being improved by us in order to provide solutions for more advanced questions and explore process enhancement options.

## Methods

**Polymerization model and algorithm**. Generally, free-radical polymerization involves at least three steps: initiation, propagation, and termination[61]. In our model, we have used a single bonding algorithm set with appropriate parameters to model radical polymerization and growth of the 3D polymer network during the DLW process. The creation of bonds is ruled by the following chemical equation of A (active) and M (monomer) particles in a given interaction range which react with the probability (ρ) and convert to polymerized monomers (T and P), if they meet criteria according to the reaction scheme shown in Fig. 7. Through initiation reaction, the monomer is transformed into an active center that can initiate polymerization reactions. In the propagation reaction, the active centers react with monomer molecules to form the first active adduct that is capable of being polymerized. Polymerizations continue in the same manner resulting in the formation of branched macromolecules that are actually end-active polymers. The final reaction is termination that can be either by oxygen quenching which deactivates the growth centers or by reaction between two polymers bearing active centers. The described procedure is implemented in our simulations by controlling the monomer type during each elementary reaction and applying the respective probability for each reaction.

The laser power influences the two-photon absorption and polymerization process by the power of two according to the Leatherdale equation for tri-acrylate photoresists[62,63]. Accordingly, in the present model, the rate of initiation reaction is influenced by the laser power and therefore the active center generation rate largely varies for different laser powers. A coefficient called initiation efficiency (α) was applied (according to Table 1) from the numerical model developed by Mueller et al.[33] to study the polymerization reaction kinetics. All relevant data and parameters employed in our simulations are reported in Table 1.

In the experiment, the nonlinear dependence of two-photon absorption (TPA) on the intensity of light, together with the relatively small TPA cross-section of materials, results in a spatial confinement of the excitation which restrict the polymerization reaction to the focal volume of a high-intensity laser called voxel[2].

In our model, the size and shape of the polymerized volume (voxel) is defined by the prolate spheroidal volume being moved three-dimensionally through the monomer pool allowing the fabrication of any 3D polymer network (shown in Fig. 8). The z axis is taken as the direction of propagation of the laser within the sample. The voxel is not moving in a continuous manner inside the monomer pool but resembling the experimental setup, the laser is applied via discrete pulses having about 20% overlap with each other. The polymerization reactions inside the voxels will happen according to the details described in the next section. Time evolution of fabrication of a sample polymer network with pyramidal shape using two different voxel sizes is shown in Fig. 8 and as a Supplementary Movie S1 in the supporting information. By scanning the laser focus through, the photoresist in three dimensions, the desired 3D architecture is defined, including the possibility of overhanging structures.

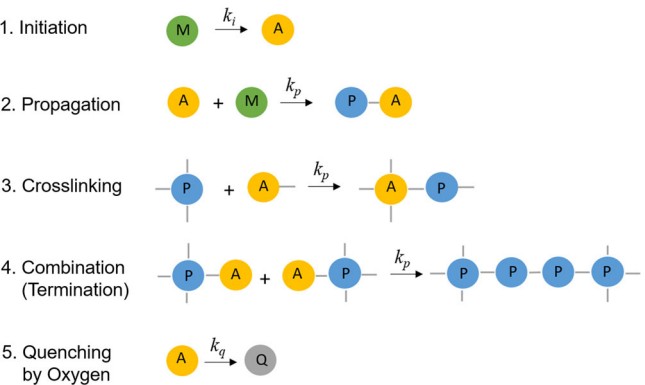

**Fig. 7 Elementary polymerization reactions implemented in our model.**
The activated particles are shown as yellow beads, free monomers (M), and monomers forming polymer chains (P) are shown in green and blue respectively. Each particle can form up to six bonds in reaction number 3.

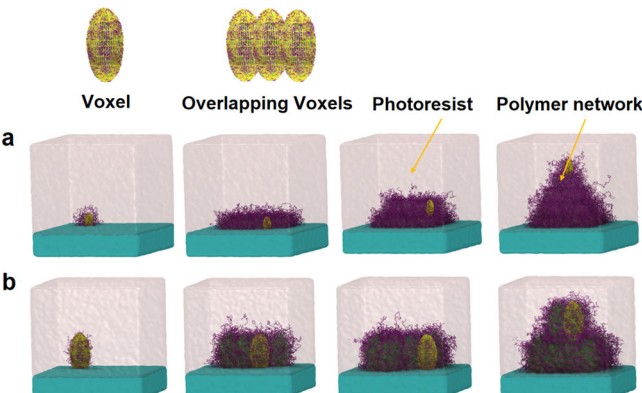

**Fig. 8 Time evolution of the pyramid growth during direct laser writing.**
The yellow ellipsoidal volume represents the laser exposure volume (voxel) and overlapping voxels allow the fabrication of polymer networks with different geometries. **a** Pyramid fabrication using small voxel size leading to a more defined structure. **b** Pyramid fabrication using larger voxel size leading to a hazy structure (Pyramid printing on 8 CPUs and 1 GPU with about 640 voxels takes more than 2 days).

The concentration of monomers and reaction rate constants were derived from experimental data for tri-acrylate-based photoresist[33]. They are often used in DLW without additional mono or bi-functional monomers for both academic and industrial purposes. The experimental data were converted to probability parameters and are reported in Table 1. It should be noted that according to our calculations each monomer has a molecular radius of about 0.5 nm, therefore, the length scale σ in our model is representative of ~1 nm length.

In the MD simulation of direct laser writing, we use a bead and bead-spring representation of the coarse-grained monomers. The simplest model for an elastomer which can capture the essential features of a rubbery (elastomer) material at the atomic scale is the model developed by Kremer and Grest[64] in which the monomers are lumped together into spherical beads and the beads are connected through elastic springs.

**Simulation details.** In the MD simulation of direct laser writing, we assume that all the particles in the system have identical masses (m) and interact via a truncated Lennard–Jones 6–12 potential equilibrated as the polymer melts at a number density $\rho = 0.85$[65]. The ε and σ parameters are the same for all monomers and set to unity in all simulations. The cutoff distance for the bead-bead interactions was set to $r_{cut} = 2.5\sigma$. The connectivity of the network mesh and beads into polymer chains was maintained by the finite extensible nonlinear elastic (FENE) potential with the spring constant $k_{spring} = 30\ k_B T/\sigma^2$ and the maximum bond length $R_{max} = 1.5\sigma$[64]. The repulsive part is represented by truncated-shifted Lennard–Jones (LJ) potential with $r_{cut} = 2^{1/6}\ \sigma$ and $\varepsilon_{LJ} = 1.5\ k_B T$

$$U_{FENE}(r) = \frac{-1}{2} k_{spring} R_{max}^2 \ln\left(1 - \frac{r^2}{R_{max}^2}\right) \qquad (1)$$

This regime is well investigated through previous simulations on different properties of polymeric materials[66–72]. We include some conditions in our simulation in order to control the polymerization reaction. The monomeric units are of functionality six, which means that during the polymerization process (resin curing); each active monomer can form up to six bonds with its neighbors. A new bond could be added to an already connected bead as long as the number of bonds per bead is less than the maximum possible number. A monomer could react with an active monomer within a certain probability if the distance of the selected bead is smaller than 1.15σ (reaction radius) from an active monomer. A modified version of bond formation algorithm implemented in LAMMPS[73] (fix bond/create), have been applied in order to allow the formation of multiple types of the bonds at the same time according to the reaction scheme illustrated earlier[40]. Reactions are performed every Θ MD steps of time step Δτ (see Table 1). In all MD simulations for polymer network formation, the NVE ensemble was adopted and periodic boundary conditions were applied on three dimensions. The number of particles in the simulation box varies from ~52,000 for cubic simulation boxes to ~1,200,000 for nano-rods that are printed voxel by voxel. All MD runs were performed using the large-scale atomic/ molecular massively parallel simulator (LAMMPS) software developed by Sandia National Laboratories[73]. After fabricating the 3D polymer networks, they were relaxed for at least $5 \times 10^4\ \tau_{LJ}$ in NPT ensemble. The temperature is maintained to be constant by coupling the system to a Langevin thermostat[74].

To calculate Young's modulus, a set of uniaxial tensile deformations were performed to obtain the stress–strain curves. The uniaxial deformation is realized by stepwise stretching the simulation box with $\Delta t = 0.01\tau$ along the x axis and at the same time compressing the box along $L_z$ and $L_y$ appropriately to maintain the simulation box being constant. We repeated the procedure and stretched the simulation box along other directions ($L_z$ and $L_y$) to obtain the mechanical response to uniaxial deformation. Very small variation has been observed between calculations which indicates that the fabricated polymer networks have homogeneous structures. A constant engineering strain rate of $v = 0.005\ \tau^{-1}$ was employed during the tensile process which caused the box dimension to change linearly with time in one dimension. It has been shown that the strain rate will affect the stress and hence calculated Young's modulus[75], but since studying this effect is not the purpose of this study, we have used a strain rate which has been used frequently for the similar coarse-grained simulations[71,76]. The applied strain rate is comparable with the segmental relaxation and practical deformation process of elastomers. The tensile stress σ in the z direction is calculated from the deviatoric

**Table 1 Parameters employed in the simulations to control laser writing condition and polymerization reactions[33].**

| Parameters | Value | Description |
|---|---|---|
| A, M, T, P, Q | - | Types of reacting (A and M) and resulting (P and T) CG beads. |
| $P_{polymerization}$ | 0.74 | The probability with which polymerization happens. |
| $P_{quenching}$ | 0.26 | The probability with which active monomers deactivate (oxygen quenching). |
| [ρ] | 0.85 | Number density of the monomers in the simulation box[33]. |
| $P_{laser}$ | $\alpha P^2$ P = laser power and α = 0.007 | The probability with which active monomers are generated[33]. |
| E | Simulation time | Exposure time ($\tau_{LJ}$) |
| Θ | 100 | Time intervals between two reactive MD steps. |

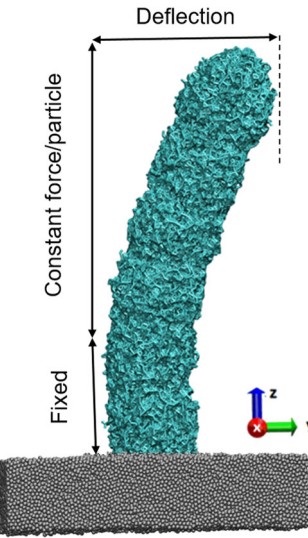

**Fig. 9** Schematic presentation of rod deformation by an external applied force and definition of deflection.

part of the stress tensor[77]:

$$\sigma = (1 + \mu)(-P_{zz} + P) = 3\frac{(-P_{zz} + P)}{2} \quad (2)$$

Where µ is the Poisson's ratio and set to be 0.5 according to the literature and $P = \sum_i P_{ii}/3$ is the hydrostatic pressure[78]. To investigate the influence of aspect ratio of the fabricated polymer network on its mechanical properties, rods with cross-section diameters $10\sigma$, $20\sigma$, $30\sigma$, and $40\sigma$ and height of $100\sigma$ were fabricated by applying laser powers ranging from 10 to 50 mW (16 rods were fabricated). Simulations were performed in a big simulation box $80\sigma \times 80\sigma \times 240\ \sigma$ (1,200,000 monomers) and rods were printed using six layers (voxels) on a substrate consisting of the fixed particles (Supplementary Movie S2). The dimension of each voxel is $\sim 48\sigma \times 24\sigma \times 24\sigma$ and each voxel contains on average 20,000 monomers. The exposure time for each voxel has been set to be 20 $\tau_{LJ}$ (printing of each rod on 8 CPUs and 1 GPU takes about 90 min). As the final preparation step, unreacted monomers were removed from the simulation box and the fabricated structures were cooled down at $T = 0.5$ for $5 \times 10^4$ timesteps since equilibration at high temperatures causes flow and a melting of the printed structures.

To investigate the stiffness of the printed structures and elucidate how they behave under an external force, we have developed a model to perform bending simulation. To bend the structure, one end of the rod is kept fixed during the simulation, which means that all degrees of freedom are set to zero for the fixed particles. A constant force per particle (F/particle), 0.004 ε/σ is applied in the $y$ direction to the unfixed part of the rods for $5 \times 10^4\ \tau_{LJ}$, letting them exert large deflection as shown in the Fig. 9. The MD simulation of deformation has been performed at $T = 0.5$ and $\Delta\tau = 0.01$ were applied. The maximum deflection of each rod has been calculated from the average atomic displacement of the particles.

We need a measure that allows us to identify local atomic events within the deformed rods. Falk and Langer[79] developed a method for determining the local deformation of an atomic system by introducing minimum non-affine squared displacement $D_{2min}$. Based on the finite-strain theory, the strain for atom i is calculated from the atomic-level deformation gradient and the strain tensor at each particle from the relative motion of its neighbors. Accordingly, the atomic strain which is measured using the Green-Lagrangian strain tensor is a good measure of local inelastic deformation[80]. All analysis and calculation of atomic strains and displacements have been performed using OVITO[60].

## Data availability

The data generated in this study along with the corresponding lammps codes have been deposited in the Zenodo database under accession code 5986853.

## Code availability

The printing protocol is available upon request under SIMSTACK workflow engine developed in our group.

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

## Acknowledgements

The authors thank Dr. Zilu Wang from the University of North Carolina at Chapel Hill, for valuable discussions on polymerization simulation techniques. We also thank Prof. Eva Blasco, Prof. Martin Wegener, Oscar Guerrero, and Pascal Kiefer for the fruitful discussion on 3D printing techniques. This research was funded by the Deutsche Forschungsgemeinschaft (DFG, German Research Foundation) under Germany's Excellence Strategy via the Excellence Cluster 3D Matter Made to Order (EXC-2082/1-390761711), by the Carl-Zeiss Foundation through the "Carl-Zeiss-Focus@HEiKA,"by the Helmholtz program "Science and Technology of Nanosystems".

## Author contributions

W.W. and E.S. conceived the study. E.S and M.L. developed the model and performed the simulations and analysis. E.S. wrote the manuscript. Supervision, project administration, and funding acquisition were done by W.W. All authors edited the manuscript and approved its final version.

## Funding

## Competing interests

The authors declare no competing interests.
