## [Peer Review File · Nature Communications]

In **Challenges and limits of Mechanical Stability in 3D Direct Laser Writing**, the authors propose a coarse-grained MD simulation to study photo-induced cross linking, and the resulting mechanical properties during laser writing. Reading the title and abstract, I was very much looking forward to reading the paper. After all, there are lots of interesting physics in the problem, and potentially useful applications. There is propagation of radiation through a material that is changing dynamically. There is a complex reaction dynamics that is aided by the material/radiation interactions. The dynamics of the chains are entangled, modified by the cross linking, interesting and complex. After the reaction is complete there could be interesting information about the mechanical properties to be probed.

Unfortunately, I was disappointed by the implementation proposed here, on a few fronts. Most of these physics were not really included, or included in a rather phenomenological way. For example, the Kremer-Grest model was used for the polymers. This model is a fairly coarse-grained model that uses significant simplifications in order to make calculations more rapid, but still might capture the universal aspects of entangled polymer dynamics, where the chains are uncrossable. But since KG uses FENE bonds, it will not make reasonable mechanical predictions on the short length scales here. Since the beads are actually many atoms lumped together, it will not have any realistic quantum effects for reaction dynamics. It's dynamics are artifactually sped up.

Moreover, the Kremer-Grest model is the most detailed variable set employed in the simulation. The radiative parts are not really there. Instead, an ellipsoidal shape is moved throughout the material in a prescribed manner, and then a continuum-level reaction dynamics is assumed inside this zone. It appears that this mass-action reaction model is just fit from experiments, if I understood the description correctly.

The justification for their approach is that existing models (alternatively and contradictorally called *analytic* and *numeric*) lack “molecular scale information and parameters that must be fitted to experiment.” But it seems to me that they are not really doing any better with regards to parameters. And because of the reasons mentioned above, I am not sure that they have any reliable molecular conformations either. In fact, given the more continuum-like aspects of their simulation, it may actually be a disadvantage to another approach that consistently uses a continuum approach. Such an approach could be done more self consistently, and more economically without losing any information found in the approach here. To be fair, there are not really sufficient details here to be sure about all that they are doing, or to reproduce the work.

The results are all qualitatively reasonable, but one probably did not need an

expensive simulation to make the same predictions. In the introduction, the authors claim the potential to provide more information than experiments, but this puts far greater faith in the simulation than I would have. Even if this simulation were performed brute force with solving Maxwell's equations, and a fully atomistic MD simulation with state-of-the-art force fields, the results would still have significant error, as nearly all demanding tests of simulation show. Nonetheless, such an approach would still pave the way for an interesting, more reliable tool, for the right kinds of questions. Of course, it would still be limited to very small sizes, like here, from computational costs. Even more interesting would be a clever use of thermodynamically and self-consistent coarse graining for such a simulation. Since they are using readily available tools like KG and LAMMPS, there is probably nothing fundamentally wrong in what they do. I just don't find the approach very novel or creative.

Review of manuscript

Challenges and limits of Mechanical Stability in 3D Direct Laser Writing

by Elaheh Sedghamiz, Modan Liu and Wolfgang Wenzel

This manuscript presents molecular dynamics simulation of direct laser writing of polymer networks. This is an interesting and technologically relevant problem. However, the manuscript lacks clarity and I have some doubts about the specifics of the implementation. Below I provide some details. In my opinion, it is not suitable for publication on Nature Communication.

At the end of the Introduction, the authors state that they “focus on an experimentally important system”, but they do not let us know what this system is. They use a generic coarse-grained model to describe the monomers undergoing polymerization. The model is derived from the bead-and-spring polymer model by Grest-Kremer, but it is supplemented with the possibility to form new bonds according to a kinetic scheme. The scheme would benefit from an illustration of the typical molecules and reactions that are being simulated (even if the coarse-grained model cannot reproduce them faithfully).

On page 5, it is written: “Fig. 2 shows the kinetics of monomer consumption in one voxel for two different exposure times, i.e. 50 and 500 time steps. For the short exposure time, the polymerization reaction occurs from 0 to 200 τ while for the longer exposure time it occurs between 0 to 170 τ .” This is not very clear to me. Also, in the Grest-Kremer model, the (Langevin) equations of motion are integrated with a time step of 0.01 τ (or lower), but the authors seem to confuse the time step and the LJ time unit.

In order to understand the effect of irradiation, which creates active monomers that may diffuse through the medium before undergoing reaction, it would be important to characterize the diffusion properties of these monomers (how far do they travel before they react?). This information seems very important, as it determines the minimum size of the features that could be produced by the technique. However, such information is completely lacking.

The authors express most quantities in reduced Lennard-Jones units ($\epsilon=1$, $\sigma=1$, $m=1$ for the beads), but the power of the laser beam starting the polymerization is given in SI units (from 10 to 50 mW). This mixing of units is confusing and should be avoided. For example, in Figure 2 the exposure time is in τ units (where τ is the LJ time) whereas power is in mW. For the same reason, sentences such as “it is sufficient to apply moderate laser powers of 40-45 mW to reach a high monomer conversion ratio” do not mean much, in the context of this model. The laser power should be given in ϵ/τ units, or alternatively ϵ , σ , m and τ should be in SI units. In connection with the laser power, where does the $\alpha=0.007$ factor come from (in Table 1)?

On page 6, it is written: “To study the mechanical properties, we fabricate cubic polymer networks applying different laser powers and constant writing velocities”. Again, this is not clear. What are the “cubes” that make up the network?

In the “Polymerization model and algorithm” section, the authors do not provide the values of the rate constants k that control the polymerization. The probabilities in Table 1 act as a substitute of the rate constants, but the text does not describe in sufficient detail how these probabilities are employed in the creation of new bonds. Also, what is the value of Θ (number of MD steps between reactions), introduced on page 13?

At the beginning of the same section, a voxel is described as a “isosurface”. To me, a voxel is a little cubic region, so I do not understand its description as a surface. What is the size of a voxel? How many polymer beads does it contain, on average?

The standard Grest-Kremer bead density of 0.85 (in LJ units, see Table 1) is appropriate for systems with purely repulsive bead-bead interactions (WCA potential, truncated at $\sigma=1.1225$). However the authors include the attractive part of the LJ potential (cutoff at $\sigma=2.5$). In my experience, this leads to instabilities, with strong heterogeneities in density (“voids”). A more reasonable value of the density of beads with attractive interaction should be between 0.95 and 1.05.

Reviewers' comments:

Reviewer #1

- What are the noteworthy results?

This paper by Sedghamiz et al. presents a two-part simulation model. The first part considers the formation of 3D polymeric networks using a simplified model for laser writing. The second part models the mechanical properties (response to extensional strain) of the resulting networks. They use a "Kremer-Grest" coarse-grained MD model for the polymer. As a consequence they are able to study the relationship between writing conditions and mechanical properties, and explore ideas like optimal writing conditions. Besides bulk 3D networks they also study structures with high-aspect ratio by probing a mechanical property that is analogous to local bending rigidity. The paper is well-organized and very readable.

- Will the work be of significance to the field and related fields? How does it compare to the established literature? If the work is not original, please provide relevant references.

While models for 3D direct laser writing have been reported [and cited in the paper], I don't know of molecular simulations that seek to model the printing process. Therein (step 1) lies the principal methodological novelty of the contribution.

- Does the work support the conclusions and claims, or is additional evidence needed?

One aspect that is not sufficiently emphasized in the current paper is the significance or additional insight that one obtains from this level of microscopic modeling. Even using qualitative arguments, most of the conclusions of the paper could be anticipated a priori. Perhaps, the authors could challenge this claim, and present their novel findings/insights (as opposed to method) to make the paper strong.

We are part of an excellence cluster on 3D printing, where one of the main challenges is to print smaller structures. For some reasons this is not working despite nearly a decade of effort and literally thousands of new initiators, polymers, printing conditions tested. We speculate that a molecular level model is required to gain insight into these phenomena and the work reported here is the first initial step in this direction. There are also a plethora of other questions we will be able to address with this model, such as multi-material printing, quenching processes, etc. However, not all of this can fit in one manuscript. We noted that Nature posted this manuscript on their own preprint server, where it was downloaded/viewed over 100 times in the first two weeks. We are therefore very optimistic that it will find a wide audience once published.

This is particularly important because no quantitative comparisons are made to experiments.

Thank you very much for your comment. We totally agree that quantitative validation can improve the manuscript and it has been added accordingly:

The experimental data concerning the structural and mechanical properties of polymer networks fabricated from the tri-acrylate family of resins via a DLW process are scarce in the literature. There exist two studies that systematically investigate the effect of laser power on the properties of the polymer networks fabricated via a DLW process. The corresponding experimental extracted from two different studies has been compared to our simulation results shown in Fig.2 (B) and Fig.3 (C).

Figure 1 shows that while the experimental degree of conversions is 5-15 % smaller than those obtained from the simulation, the overall trend is very close to the experiment, especially at very low and very high laser powers.

For the mechanical properties, experimentally it has been observed that the progression of Young's modulus does not show a saturation behavior as the double-bond conversion does. While the simulation also shows a linear behavior for the laser powers below 30 mW, Young's modulus saturates at high laser powers.

The discrepancy between the experimental and simulation trend for the mechanical properties is related to the fact that the existing experimental measurements of Young's modulus are limited to the low laser powers below 30 mW due to some technical problems in measuring Young's modulus of structures fabricated with higher laser powers. However, there is no limitation in applying higher laser powers in the simulations and while for low laser powers the simulation results are consistent with that of the experiment, we observed that the relationship between the laser power and Young's modulus is not linear but shows a second-order behavior similar to that of the degree of monomer conversion.

The discussion mentioned above has been added to the manuscript.

Thus, the question "how do we know the model is correct or useful?" is not addressed. Yes, the authors describe an approximate 2-step protocol to model the problem that seems reasonable. Qualitatively reasonable conclusions flow from it. But our trust in the model would be higher if there were external quantitative validation, or if it could shed light on some puzzling feature.

- Are there any flaws in the data analysis, interpretation and conclusions? - Do these prohibit publication or require revision?

I have some questions about the first step of the simulation [below], but these may be addressable. My main complaint is the one I lodged above regarding validation of the model. In my opinion, the model presented ought to (a) **(qualitatively) shed light on some unresolved mystery, or (b) (quantitatively) be compared with experiments**. This paper, while extremely interesting, doesn't meet that threshold. I do believe bolstering the paper along any of these two directions would make a much stronger case.

We hope we can answer both of these points: the unsolved mystery we want to address is whether molecular scale effects become relevant in the mechanical properties of 3D printed systems when lowering the resolution. Already at resolutions below 400nm the stability of printed structures becomes problematic. We have pointed out above that printing below 100n, is a major challenge that is presently not met. This is NOT because the STED voxel cannot be focused sufficiently, but because no stable structures form. There are no imaging methods available to shed light on these questions. In principle, there are 2 scenarios: 1) the reaction breaks down at small length scales or 2) the structures that form are so unstable they dissolve/break during processing. We see some evidence for scenario (2) but only at very

small scales. We agree that it is important to compare with experiment and we have improved the manuscript as discussed above.

- Is the methodology sound? Does the work meet the expected standards in your field?

The modeling of the formation of the 3D networks is somewhat crude. If I understand it correctly, they perform a regular MD simulation, and every τ_R steps scan the local neighborhood to form new bonds according to the probabilities obtained from kinetics. I did not quite understand why $\tau_R = 10 \tau$?

Thank you very much for your comment.

To refer briefly, in the bond-generating algorithm, each active monomer selects a monomer bead to bond within a neighboring sphere of radius R within the probability ρ . This neighbor becomes an active monomer and the previous active site becomes part of the chain. After each reactive step, the system is allowed to relax during $t=0.1\tau$ within the NVE ensemble. The relaxation time between two reactive steps (Θ) is considered as a part of the standard procedure for this type of simulation (J. Chem. Phys. **142**, 134102 (2015), ACS *omega*, **3**, 15426 (2018)). The choice of $\Theta=10$ is somewhat arbitrary and different Θ values have been applied in different studies. We also examined different Θ values (10, 20, and 100) and no significant difference was observed between the selected values. Therefore $\Theta=10$ was applied in all simulations.

Furthermore, the effect of local thermal heating, which should affect reaction rates and affect particle trajectories is mostly neglected. This may be especially relevant at high powers.

Thanks for referring to this important issue. Throughout this project, we were working in close interaction with experimentalists experts in the field of direct laser writing (within the cluster of excellence 3D matter made to order). The following is according to their observations which have been published only partly and the rest are discussed in the Ph.D. thesis of Jonathan Mueller:

They have performed in-situ local temperature measurement in the exposed volume during three-dimensional direct laser writing. It has been observed that the change in temperature is not more than approximately 5 K, even if the writing power is increased significantly above the writing threshold (Appl. Phys. Lett. 103, 123107 (2013)).

“Generally, from their experiments, it is quite clear that thermal effects play no or only a minor role for DLW. From the polymer chemistry side, one would expect to see an increase in the polymerization velocity for higher temperatures due to the higher mobility of the reagents. However, this increase in mobility would affect not only the chain-growth mechanism but also the quenching reaction (oxygen) in a similar way, so that at first order, the overall effect could be expected to cancel out.

According to their study, the dominant mechanism for radical formation is purely photochemical at all investigated repetition rates, while an (at least partly) photo-thermal

process is connected to the sample damage in the overexposure regime. This observation is in perfect agreement with the behavior of the photo-resist with respect to the laser repetition rate”

According to the above observations, we did not take into account any thermal effect and assumed that the high laser powers will not affect the reaction rates and polymerization mechanism.

- Is there enough detail provided in the methods for the work to be reproduced?

I think so.

Reviewer 2

In Challenges and limits of Mechanical Stability in 3D Direct Laser Writing, the authors propose a coarse-grained MD simulation to study photo-induced cross linking, and the resulting mechanical properties during laser writing. Reading the title and abstract, I was very much looking forward to reading the paper. After all, there are lots of interesting physics in the problem, and potentially useful applications. There is propagation of radiation through a material that is changing dynamically. There is a complex reaction dynamics that is aided by the material/radiation interactions. The dynamics of the chains are entangled, modified by the cross linking, interesting and complex. After the reaction is complete there could be interesting information about the mechanical properties to be probed. Unfortunately, I was disappointed by the implementation proposed here, on a few fronts. Most of these physics were not really included, or included in a rather phenomenological way. For example, the Kremer-Grest model was used for the polymers. This model is a fairly coarse-grained model that uses significant simplifications in order to make calculations more rapid, but still might capture the universal aspects of entangled polymer dynamics, where the chains are uncrossable. But since KG uses FENE bonds, it will not make reasonable mechanical predictions on the short length scales here. Since the beads are actually many atoms lumped together, it will not have any realistic quantum effects for reaction dynamics. It's dynamics are artifactually sped up.

Thank you very much for your comments.

We agree that due to the asymmetric behavior of FENE bond potential, it is expected that the long bonds are more favorable than short bonds, which is a disadvantage in simulations where the elongation of the bonds is considered. It has been shown that this drawback can be overcome by applying stiff FENE+LJ potential ($k=30$ and $R_0= 1.5s$) guaranteeing a certain stiffness of the bonds while avoiding high-frequency modes and chain crossing. (*ACS Macro Lett.* 2018, 7, 1, 116–121, *Phys. Chem. Chem. Phys.*, 2018, **20**, 8228-8240). We want to point out that we use a form of the potential that avoids these problems.

Such a stiff FENE+LJ potential has been shown to produce mechanical properties with small deviations from the experiment. Using the same potential, Wei et al. (*Phys. Chem. Chem. Phys.*, 2018, **20**, 8228-8240) have shown that the mechanical properties of natural rubber based on the Kremer-Grest polymer model is not mainly determined by the intramolecular interaction but by the intermolecular interaction. They also showed that the spring constant of FENE bonds has the ability to restrict the bond stretching, and thereby influences the mechanical properties in terms of conformational entropy and provides a more physical behavior. We believe that the potential adopted here can at least qualitatively reflect the mechanical behavior of real polymer networks.

Moreover, the Kremer-Grest model is the most detailed variable set employed in the simulation. The radiative parts are not really there. Instead, an ellipsoidal shape is moved throughout the material in a prescribed manner, and then a continuum-level reaction dynamics is assumed inside this zone. It appears that this mass-action reaction model is just fit from experiments, if I understood the description correctly.

The radiative part of the simulation is governed by an ellipsoidal volume (voxel) in which the polymerization reactions are allowed to happen. The way that this voxel moves inside the monomer pool completely corresponds to the movement of the laser in the experimental setup where the voxel does not move in a continuous manner, but discrete voxels are placed in close proximity so as to induce an overlap of about 20% (similar to the image below). This behavior is modelled by an in-house code that is synced with the LAMMPS package. Parameters that are used in the radiative part stem from an established numerical model of Mueller et al. (*Advanced Materials*, 2014, **26**(38), 6566-71) which was developed to reproduce the time-dependent concentrations of the reacting species at the threshold writing laser power.

To clarify the above-mentioned procedure to the readers, we have added more information on the radiative part to the manuscript.

The justification for their approach is that existing models (alternatively and contradictorally called analytic and numeric) lack “molecular scale information and parameters that must be fitted to experiment.” But it seems to me that they are not really doing any better with regards to parameters. And because of the reasons mentioned above, I am not sure that they have any reliable molecular conformations either. In fact, given the more continuum-like aspects of their simulation, it may actually be a disadvantage to another approach that consistently uses a continuum approach. Such an approach could be done more self consistently, and more economically without losing any information found in the approach here.

As discussed in detail in the response to referee 1 we have now explicitly included quantitative comparison to experiment. We understand the concern of the referee which stems in part from the observation that it is difficult to characterize the printed structures at a scale of less than 100 nm. We believe that this is all the more reason to pursue such molecular models and we hope that the referee can consider this in his/her decision. If we ever want to understand the impact of the choice of the molecules on the printing process and the quality of the result, we need to model or measure at the molecular scale. In this sense, the present work represents a breakthrough that makes simulations with molecular resolution on the 100 nm scale feasible for the first time.

To be fair, there are not really sufficient details here to be sure about all that they are doing, or to reproduce the work. The results are all qualitatively reasonable, but one probably did not need an expensive simulation to make the same predictions. In the introduction, the authors claim the potential to provide more information than experiments, but this puts far greater faith in the simulation than I would have. Even if this simulation were performed brute force with solving Maxwell's equations, and a fully atomistic MD simulation with state-of-the-art force fields, the results would still have significant error, as nearly all demanding tests of simulation show. Nonetheless, such an approach would still pave the way for an interesting, more reliable tool, for the right kinds of questions. Of course, it would still be limited to very small sizes, like here, from computational costs. Even more interesting would be a clever use of thermodynamically and self-consistent coarse graining for such a simulation. Since they are using readily available tools like KG and LAMMPS, there is probably nothing fundamentally wrong in what they do. I just don't find the approach very novel or creative.

Of course, continuum mechanical descriptions of materials behavior can be profitably applied for many engineering questions connected to the mechanical properties of materials. However, continuum methods require constitutive equations and the required parameters. Both are completely lacking in this field and we know from many other applications that the derivation and calibrations of these models is a very complex and challenging undertaking.

Moreover, the mechanical and physical properties of polymeric materials originate from the interplay of phenomena at different spatial and temporal scales. To understand the relationship between the morphology (e.g degree of monomer conversion) and mechanical properties of a polymer network it is usually indispensable to apply a molecular scale approach, as macroscopic properties will be ultimately determined by events on the molecular scale. Molecular scale properties are essentially impossible to measure in present-day experiments and the data concerning this relationship are very scarce in the literature.

While continuum models are unavailable because their parameters are unknown, all-atom MD is clearly not the solution. The simulations presented in this paper would require years of CPU wallclock time using an established atomistic forcefield and it's anybody's guess whether the established forcefields would be any better. We work as a group in molecular electronics for more than a decade and in our experience this is often not the case. The idea of systematic coarse graining suggested by the referee is indeed an interesting one, yet it would require some reasonable approach to model the system. Alas, this paper should be published, because there is none at the moment. To pursue thermodynamic

coarse graining, one should also have an idea how the non-self-consistent approach performs. These are, we believe, strong reasons to publish this manuscript.

Regarding the morphologies of the polymer networks, our coarse-grained representation focuses on the monomer kinetics by considering the interplay of the monomer diffusion, and the chemical reactions. There the competing entropic placement of the monomers and the probability-based enthalpic addition of constraints comprehensively regulate the conformation of the printed polymer network. Here, the characteristic timescale is at least one order of magnitude slower than the activation reaction and the termination reactions involved in DLW.

We are aware that the current approach is merely a first step, but without a first step, there simply is no second step. Surely, some deficiencies exist which are connected to the simplification of the applied methodology (Kremer-Grest polymer model) and limitations inherent to the MD approach. Nevertheless, we believe that as the first molecular dynamic study of direct laser writing and as a fast and affordable computational approach, the current study can serve to better understand the DLW process and determine optimal process settings. It can pave the way for more computational studies in this field and can be further improved by us or other research groups in order to provide solutions for more advanced questions and explore process enhancement options.

Some information on the outlook has been added to the manuscript.

Reviewer 3

Challenges and limits of Mechanical Stability in 3D Direct Laser Writing by Elaheh Sedghamiz, Modan Liu and Wolfgang Wenzel. This manuscript presents molecular dynamics simulation of direct laser writing of polymer networks. This is an interesting and technologically relevant problem. However, the manuscript lacks clarity and I have some doubts about the specifics of the implementation. Below I provide some details. In my opinion, it is not suitable for publication on Nature Communication. At the end of the Introduction, the authors state that they “focus on an experimentally important system”, but they do not let us know what this system is.

By the term “experimentally important system”, we refer to direct laser writing using the tri-acrylate family of photoresists that are being widely used especially for commercial purposes. We have clarified this in the manuscript.

They use a generic coarse-grained model to describe the monomers undergoing polymerization. The model is derived from the bead-and-spring polymer model by Grest-Kremer, but it is supplemented with the possibility to form new bonds according to a kinetic scheme. **The scheme would benefit from an illustration of the typical molecules and reactions that are being simulated** (even if the coarse-grained model cannot reproduce them faithfully). On page 5, it is written: “Fig. 2 shows the kinetics of monomer consumption in one voxel for two different exposure times, i.e. 50 and 500 time steps. For the short exposure time, the

polymerization reaction occurs from 0 to 200 τ while for the longer exposure time it occurs between 0 to 170 τ ." This is not very clear to me. Also, in the Grest-Kremer model, the (Langevin) equations of motion are integrated with a time step of 0.01 τ (or lower), but the authors seem to confuse the time step and the LJ time unit.

Thank you very much for pointing this out. We apologize for this typo. All corresponding plot axes have been changed to the LJ time unit. A figure with details about the molecules and reactions has been added.

In order to understand the effect of irradiation, which creates active monomers that may diffuse through the medium before undergoing reaction, it would be important to characterize the diffusion properties of these monomers (how far do they travel before they react?). This information seems very important, as it determines the minimum size of the features that could be produced by the technique. However, such information is completely lacking.

Thanks for your comment. In our current approach, the effect of the irradiation results in a local conversion of active monomers, which promptly reacts with their neighboring monomers if they meet the polymerization criteria specified by the polymerization condition. Since the number density of the monomers in the resin is reasonably high at 0.8, the life-time of active monomers is $\sim 4\tau$ which is extremely short, while the mean free path is measured at $\sim 0.2\sigma$ for the initial stage of the printing, and a reduction to $\sim 0.15\sigma$ as the polymer network grows.

In lab conditions, the generation of active monomers only starts after the oxygen in the local vicinity gets depleted, whereas the timescale corresponding to the oxygen diffusion is much smaller than that of the monomers. The investigation of how oxygen diffusion regulates the minimum feature size of printing is the subject of our future study.

The above information has been added to the manuscript.

The authors express most quantities in reduced Lennard-Jones units ($\epsilon=1$, $\sigma=1$, $m=1$ for the beads), but the power of the laser beam starting the polymerization is given in SI units (from 10 to 50 mW). This mixing of units is confusing and should be avoided. For example, in Figure 2 the exposure time is in τ units (where τ is the LJ time) whereas power is in mW. For the same reason, sentences such as "it is sufficient to apply moderate laser powers of 40-45 mW to reach a high monomer conversion ratio" do not mean much, in the context of this model. The laser power should be given in ϵ/τ units, or alternatively ϵ , σ , m and τ should be in SI units.

Our current model can be considered as a generic model for the tri-acrylate families of resins. Therefore, our aim is not to provide quantitative results on the mechanical properties since there is a significant difference in the corresponding values for different tri-acrylate based resins (but the overall trend is the same). On the other hand, the performance of our model has been evaluated against some experimental data. Therefore, in all cases, the laser power is reported in the actual values in order to make it easier for the audiences that can also be experimentalists to evaluate the model.

In connection with the laser power, where does the $\alpha=0.007$ factor come from (in Table 1)?

Most of the parameters on the radiative and polymerization part of the simulations are borrowed from the numerical model developed to study the polymerization reaction kinetics in direct laser writing by Mueller et al. (Advanced Materials. 2014, 26(38), 6566-71 which has been already cited in the manuscript). The numerical model uses both experimental and fitted data. The initiation efficiency ($\alpha=0.007$) is also from the numerical model which reproduces the time-dependent concentrations of the reacting species at the threshold writing laser power. To clarify this also for the readers, we have added more information on this parameter to the manuscript.

On page 6, it is written: "To study the mechanical properties, we fabricate cubic polymer networks applying different laser powers and constant writing velocities". Again, this is not clear. What are the "cubes" that make up the network?

The cubic polymer networks presented in the first part of the study are representative of one voxel (these cubes are not fabricated voxel by voxel). The reason is that in the first part of the study, our aim is to study the relationship between the laser power (or exposure time) and the degree of monomer conversion. The corresponding data have been used to validate the behavior of our model against the experimental data; therefore, it is reasonable to study the behavior of only one voxel to avoid the effect of voxel specifications (voxel overlapping, voxel size, shape, etc...) on the degree of monomer conversion.

In the "Polymerization model and algorithm" section, the authors do not provide the values of the rate constants k that control the polymerization. The probabilities in Table 1 act as a substitute of the rate constants, but the text does not describe in sufficient detail how these probabilities are employed in the creation of new bonds. Also, what is the value of Θ (number of MD steps between reactions), introduced on page 13?

Thank you very much for pointing this out. We apologize for this inconsistency. The value of $\Theta=10$ was reported as τ_R in table 1. The τ_R symbol has been changed to Θ to be unified through the text and table.

Since the method of bond creation by employing the corresponding probabilities is part of the standard "fix bond/create command" implemented in the LAMMPS simulation package, we did not add more explanation on this part to the manuscript as the corresponding information is documented on the LAMMPS official website (https://lammmps.sandia.gov/doc/fix_bond_create.html) under the "fix bond/create" keyword.

At the beginning of the same section, a voxel is described as a "isosurface". To me, a voxel is a little cubic region, so I do not understand its description as a surface. What is the size of a voxel? How many polymer beads does it contain, on average?

The definition of the voxel has been revised.

Inspired from the experimental shape of the voxel, in our model, the voxel is not a cubic region but is a prolate spheroidal volume being moved three-dimensionally through the box of free monomers allowing the formation of polymer networks and generating the nano-rods as is also shown in the supplementary movies.

The voxel size is $(48 \times 24 \times 24) \sigma$ and the average number of monomers inside the voxel is about 20000

The above information has been added to the method section.

The standard Grest-Kremer bead density of 0.85 (in LJ units, see Table 1) is appropriate for systems with purely repulsive bead-bead interactions (WCA potential, truncated at $\sigma=1.1225$). However the authors include the attractive part of the LJ potential (cutoff at $\sigma=2.5$). In my experience, this leads to instabilities, with strong heterogeneities in density ("voids"). A more reasonable value of the density of beads with attractive interaction should be between 0.95 and 1.05.

Thank you very much for the information. The number density 0.85 was chosen to be compatible with the experimental number density of monomers inside the monomer pool.

REVIEWER COMMENTS

Reviewer #1 (Remarks to the Author):

1. The authors addressed my question about validating the model with experiments. Thank you for that.
2. The proposed computational model is phenomenological with quite a few ad hoc elements. It may still be useful, but my confidence in it is quite a bit lower than the authors'.
3. The paper is publishable because it fills a void in the literature. I am not sure if it is suitable for Nature Comm. due to the crudeness of the model.
4. I am sympathetic to the idea that this is a first step. However, like Reviewer 2, I don't think the computational method is sufficiently reliable or innovative.

Taken together, I strongly suggest that the authors publish in a high-quality/workhorse journal.

Reviewer #2 (Remarks to the Author):

The response of the authors to the comments are quite reasonable and appropriate. I have large agreements with the concerns of the first reviewer also, and the authors' responses there did clarify several things. And in their response to my comment, that this is just a first step, necessary for a second step is completely correct, in my opinion.

I have only tiny complaints about their conclusions. So, it is the odd state that we are in large agreement about the technical details, but in disagreement about the magnitude of the results. Unfortunately, I still find the approach to be not very novel, and just what is expected as the easiest way (though maybe not completely trivial) to tackle this problem. Similarly, there were no surprising results to challenge our intuition. Yes, it is true that the comparison with experiments were not too bad, but that is not really a very high bar in this case.

Absolutely the paper should be published. It will probably even get significant citations. But it is not quite up to the aspirations of this journal, in my opinion. Other honest researchers might disagree, and I could respect that, but this is my opinion.

Reviewer #4 (Remarks to the Author):

Reviewer 4 responding in Reviewer 3's absence:

Reviewer 3: Challenges and limits of Mechanical Stability in 3D Direct Laser Writing by Elaheh Sedghamiz, Modan Liu and Wolfgang Wenzel. This manuscript presents molecular dynamics simulation of direct laser writing of polymer networks. This is an interesting and technologically relevant problem. However, the manuscript lacks clarity and I have some doubts about the specifics of the implementation. Below I provide some details. In my opinion, it is not suitable for publication on Nature Communication. At the end of the Introduction, the authors state that they “focus on an experimentally important system”, but they do not let us know what this system is.

Authors' response: By the term “experimentally important system”, we refer to direct laser writing using the tri-acrylate family of photoresists that are being widely used especially for commercial purposes. We have clarified this in the manuscript.

Reviewer 4: This modification was simply the addition of a clause defining the class of monomer studied. The novelty and interesting part of this work is in the simulated irradiation of the thermoset: Rastering the focal region of a laser in 3-dimensions to simulate direct laser writing by two-photon polymerization like in the commercial Nanoscribe instrument. The approach toward simulated polymerization of a trifunctional acrylate is not as novel and has been simulated at the all-atom level in recent work (J. Phys. Chem. B 2020, 124, 9204–9215. <https://doi.org/10.1021/acs.jpcc.0c05319>.) I find this work, simulation of 2PP DLW by coarse-grained MD, more impactful than Reviewers 3 and 2 do, with similar reasoning as Reviewer 1.

Reviewer 3: They use a generic coarse-grained model to describe the monomers undergoing polymerization. The model is derived from the bead-and-spring polymer model by Grest-Kremer, but it is supplemented with the possibility to form new bonds according to a kinetic scheme. The scheme would benefit from an illustration of the typical molecules and reactions that are being simulated (even if the coarse-grained model cannot reproduce them faithfully). On page 5, it is written: “Fig. 2 shows the kinetics of monomer consumption in one voxel for two different exposure times, i.e. 50 and 500 time steps. For the short exposure time, the polymerization reaction occurs from 0 to 200 τ while for the longer exposure time it occurs between 0 to 170 τ .” This is not very clear to me. Also, in the Grest-Kremer model, the (Langevin) equations of motion are integrated with a time step of 0.01 τ (or lower), but the authors seem to confuse the time step and the LJ time unit.

Authors' response: Thank you very much for pointing this out. We apologize for this typo. All corresponding plot axes have been changed to the LJ time unit. A figure with details about the molecules and reactions has been added.

Reviewer 4: The authors have fully addressed Reviewer 3's concerns.

Reviewer 3: In order to understand the effect of irradiation, which creates active monomers that may diffuse through the medium before undergoing reaction, it would be important to characterize the diffusion properties of these monomers (how far do they travel before they react?). This information seems very important, as it determines the minimum size of the features that could be produced by the technique. However, such information is completely lacking.

Authors' response: Thanks for your comment. In our current approach, the effect of the irradiation results in a local conversion of active monomers, which promptly reacts with their neighboring monomers if they meet the polymerization criteria specified by the polymerization condition. Since the number density of the monomers in the resin is reasonably high at 0.8, the life-time of active monomers is $\sim 4\tau$ which is extremely short, while the mean free path is measured at $\sim 0.2\sigma$ for the initial stage of the printing, and a reduction to $\sim 0.15\sigma$ as the polymer network grows.

In lab conditions, the generation of active monomers only starts after the oxygen in the local vicinity gets depleted, whereas the timescale corresponding to the oxygen diffusion is much smaller than that of the monomers. The investigation of how oxygen diffusion regulates the minimum feature size of printing is the subject of our future study.

The above information has been added to the manuscript.

Reviewer 4: The authors respond quantitatively to Reviewer 3's questions, but these responses highlight a potential shortcoming of this work. The $\sim 0.2\sigma$ should be put into context with estimated travel/mean free path of species in experimental conditions.

Additionally, readers (and Reviewers) would benefit from additional emphasis on limitations of computational resources, that these models and methods are a pioneering framework and that, as compute power grows, the complexity of the MD parts of this model can achieve greater fidelity. These simulations, with $1.2e6$ particles, are big MD simulations compared with many contemporary works, mention of the resources/wall times required for the large-scale simulations would be very interesting.

Reviewer 3: The authors express most quantities in reduced Lennard-Jones units ($\epsilon=1$, $\sigma=1$, $m=1$ for the beads), but the power of the laser beam starting the polymerization is given in SI units (from 10 to 50 mW). This mixing of units is confusing and should be avoided. For example, in Figure 2 the

exposure time is in τ units (where τ is the LJ time) whereas power is in mW. For the same reason, sentences such as “it is sufficient to apply moderate laser powers of 40-45 mW to reach a high monomer conversion ratio” do not mean much, in the context of this model. The laser power should be given in ϵ/τ units, or alternatively ϵ , σ , m and τ should be in SI units.

Authors’ response: Our current model can be considered as a generic model for the tri-acrylate families of resins. Therefore, our aim is not to provide quantitative results on the mechanical properties since there is a significant difference in the corresponding values for different tri-acrylate based resins (but the overall trend is the same). On the other hand, the performance of our model has been evaluated against some experimental data. Therefore, in all cases, the laser power is reported in the actual values in order to make it easier for the audiences that can also be experimentalists to evaluate the model.

Reviewer 4: I agree with the last suggestion of Reviewer 3 and believe that this work will more easily reach experimentalists, as mentioned above, by conversion to SI units. L-J (or other reduced units) will limit the engagement of the interested non-specialist and introduce an unnecessary barrier in a journal with broad readership.

Reviewer 3: In connection with the laser power, where does the $\alpha=0.007$ factor come from (in Table 1)?

Authors’ response: Most of the parameters on the radiative and polymerization part of the simulations are borrowed from the numerical model developed to study the polymerization reaction kinetics in direct laser writing by Mueller et al. (Advanced Materials. 2014, 26(38), 6566-71 which has been already cited in the manuscript). The numerical model uses both experimental and fitted data. The initiation efficiency ($\alpha=0.007$) is also from the numerical model which reproduces the time-dependent concentrations of the reacting species at the threshold writing laser power. To clarify this also for the readers, we have added more information on this parameter to the manuscript.

Reviewer 4: The authors address Reviewer 3’s question adequately.

Reviewer 3: On page 6, it is written: “To study the mechanical properties, we fabricate cubic polymer networks applying different laser powers and constant writing velocities”. Again, this is not clear. What are the “cubes” that make up the network?

Authors’ response: The cubic polymer networks presented in the first part of the study are representative of one voxel (these cubes are not fabricated voxel by voxel). The reason is that in the first part of the study, our aim is to study the relationship between the laser power (or exposure time) and the degree of monomer conversion. The corresponding data have been used to validate the behavior of our model against the experimental data; therefore, it is reasonable to study the

behavior of only one voxel to avoid the effect of voxel specifications (voxel overlapping, voxel size, shape, etc...) on the degree of monomer conversion.

Reviewer 4: The authors' text and response was not clear to me either. I assume that by "cubes" the authors used simulation boxes with uniform 'irradiation' and periodic boundary conditions in x, y, and z to simulate a bulk triacrylate thermoset, not a voxel. Is this correct? If so, stating that a "cube" represents a voxel is incorrect since the edge effects of the voxel (the thermoset/liquid interface) are ignored. It is the finite size that defines the voxel. The authors also state elsewhere that investigation of this interface, in the context of writing resolution, will be considered in future work. Therefore, I suggest that this section be revised again for readability.

Reviewer 3: In the "Polymerization model and algorithm" section, the authors do not provide the values of the rate constants k that control the polymerization. The probabilities in Table 1 act as a substitute of the rate constants, but the text does not describe in sufficient detail how these probabilities are employed in the creation of new bonds. Also, what is the value of Θ (number of MD steps between reactions), introduced on page 13?

Authors' response: Thank you very much for pointing this out. We apologize for this inconsistency. The value of $\Theta=10$ was reported as τ_R in table 1. The τ_R symbol has been changed to Θ to be unified through the text and table.

Since the method of bond creation by employing the corresponding probabilities is part of the standard "fix bond/create command" implemented in the LAMMPS simulation package, we did not add more explanation on this part to the manuscript as the corresponding information is documented on the LAMMPS official website (https://lammps.sandia.gov/doc/fix_bond_create.html) under the "fix bond/create" keyword.

Reviewer 4: To permit reproducibility, use of bond/create should be mentioned explicitly in the main text. The incorporation of the authors' parameters into that fix's implementation are then rather straightforward. The authors should also mention use of SIMSTACK and its relation to LAMMPS in the main text. SIMSTACK appears to be integral to these simulations.

Reviewer 3: At the beginning of the same section, a voxel is described as a "isosurface". To me, a voxel is a little cubic region, so I do not understand its description as a surface. What is the size of a voxel? How many polymer beads does it contain, on average?

Authors' response: The definition of the voxel has been revised.

Inspired from the experimental shape of the voxel, in our model, the voxel is not a cubic region but is a prolate spheroidal volume being moved three-dimensionally through the box of free monomers

allowing the formation of polymer networks and generating the nano-rods as is also shown in the supplementary movies.

The voxel size is $(48 \times 24 \times 24) \sigma$ and the average number of monomers inside the voxel is about 20000

The above information has been added to the method section.

Reviewer 4: Reviewer 3's confusion is understandable. Related to my earlier responses, I believe that the manuscript would be strengthened by a clearer description of the two-photon polymerization DLW instrumentation, where the voxel ("volumetric pixel") has the shape described by the authors, a result of quadratic dependence on laser power. The voxel size and monomer contents questions are answered adequately.

Reviewer 3: The standard Grest-Kremer bead density of 0.85 (in LJ units, see Table 1) is appropriate for systems with purely repulsive bead-bead interactions (WCA potential, truncated at $\sigma=1.1225$). However the authors include the attractive part of the LJ potential (cutoff at $\sigma=2.5$). In my experience, this leads to instabilities, with strong heterogeneities in density ("voids"). A more reasonable value of the density of beads with attractive interaction should be between 0.95 and 1.05.

Authors' response: Thank you very much for the information. The number density 0.85 was chosen to be compatible with the experimental number density of monomers inside the monomer pool.

Reviewer 4: There may be misunderstanding here. Reviewer 3 is suggesting that the selected number density of 0.85 can lead to instabilities and heterogeneity, not that 0.85 disagrees with experiment. To respond to this, the authors should validate the spatial homogeneity of their simulated resin, particularly considering the authors' use of the NVE ensemble.

Additional Reviewer 4 comment:

Proofreading is required. A few examples:

I believe Figure 3D and E in the main text refer to Figures 3B and C, Figure 2 refers to Figure 1 in one instance. Sentence "No high strain region can be detected for samples with." requires revision.

The addition of experimental data in Figure 3C appears to correspond to one of the two experimental data sets in Figure 4 of Reference 49. If so, the authors should explicitly clarify the choice to show one data set over the other or if some other intermediate analysis or data from this reference was used.

We would like to thank the Editor for taking the time to consider our manuscript and the Reviewers for carefully reviewing our manuscript and providing constructive suggestions. On behalf of all authors, I am submitting the revised manuscript titled “Challenges and Limits of Mechanical Stability in 3D Direct Laser Writing” by Elaheh Sedghamiz, Modan Liu and Wolfgang Wenzel in which we address the comments of the reviewers. The point-by-point response to the comments is as follows:

Reviewer #4 (Remarks to the Author):

Reviewer 4 responding in Reviewer 3's absence:

Reviewer 3: Challenges and limits of Mechanical Stability in 3D Direct Laser Writing by Elaheh Sedghamiz, Modan Liu and Wolfgang Wenzel. This manuscript presents molecular dynamics simulation of direct laser writing of polymer networks. This is an interesting and technologically relevant problem. However, the manuscript lacks clarity and I have some doubts about the specifics of the implementation. Below I provide some details. In my opinion, it is not suitable for publication on Nature Communication. At the end of the Introduction, the authors state that they “focus on an experimentally important system”, but they do not let us know what this system is.

Authors' response: By the term “experimentally important system”, we refer to direct laser writing using the tri-acrylate family of photoresists that are being widely used especially for commercial purposes. We have clarified this in the manuscript.

Reviewer 4: This modification was simply the addition of a clause defining the class of monomer studied. The novelty and interesting part of this work is in the simulated irradiation of the thermoset: Rastering the focal region of a laser in 3-dimensions to simulate direct laser writing by two-photon polymerization like in the commercial Nanoscribe instrument. The approach toward simulated polymerization of a trifunctional acrylate is not as novel and has been simulated at the all-atom level in recent work (J. Phys. Chem. B 2020, 124, 9204–9215. <https://doi.org/10.1021/acs.jpcc.0c05319>.) I find this work, simulation of 2PP DLW by coarse-grained MD, more impactful than Reviewers 3 and 2 do, with similar reasoning as Reviewer 1.

Reply: We thank the reviewer for reviewing our manuscript and pointing out the impact of our work and have cited the paper in the revision.

Reviewer 3: They use a generic coarse-grained model to describe the monomers undergoing polymerization. The model is derived from the bead-and-spring polymer model by Grest-Kremer, but it is supplemented with the possibility to form new bonds according to a kinetic scheme. The scheme would benefit from an illustration of the typical molecules and reactions that are being simulated (even if the coarse-grained model cannot reproduce them faithfully). On page 5, it is written: “Fig. 2 shows the kinetics of monomer consumption in one voxel for two different exposure times, i.e. 50 and 500 time steps. For the short exposure time, the polymerization reaction occurs from 0 to 200 τ while for the longer exposure time it occurs between 0 to 170 τ .”. This is not very clear to me. Also, in the Grest-Kremer model, the

(Langevin) equations of motion are integrated with a time step of 0.01τ (or lower), but the authors seem to confuse the time step and the LJ time unit.

Authors' response: Thank you very much for pointing this out. We apologize for this typo. All corresponding plot axes have been changed to the LJ time unit. A figure with details about the molecules and reactions has been added.

Reviewer 4: The authors have fully addressed Reviewer 3's concerns.

Reviewer 3: In order to understand the effect of irradiation, which creates active monomers that may diffuse through the medium before undergoing reaction, it would be important to characterize the diffusion properties of these monomers (how far do they travel before they react?). This information seems very important, as it determines the minimum size of the features that could be produced by the technique. However, such information is completely lacking.

Authors' response: Thanks for your comment. In our current approach, the effect of the irradiation results in a local conversion of active monomers, which promptly react with their neighboring monomers if they meet the polymerization criteria specified by the polymerization condition. Since the number density of the monomers in the resin is reasonably high at 0.8, the life-time of active monomers is $\sim 4\tau$ which is extremely short, while the mean free path is measured at $\sim 0.2\sigma$ for the initial stage of the printing, and a reduction to $\sim 0.15\sigma$ as the polymer network grows.

In lab conditions, the generation of active monomers only starts after the oxygen in the local vicinity gets depleted, whereas the timescale corresponding to the oxygen diffusion is much smaller than that of the monomers. The investigation of how oxygen diffusion regulates the minimum feature size of printing is the subject of our future study.

The above information has been added to the manuscript.

Reviewer 4: The authors respond qualitatively to Reviewer 3's questions, but these responses highlight a potential shortcoming of this work. The $\sim 0.2\sigma$ should be put into context with estimated travel/mean free path of species in experimental conditions.

Reply: Thank you very much for your comment. In our simulations 0.2σ corresponds to about 0.2 nm and the life time of the active monomers is about 40 ps which is extremely short. In agreement with experimental observations, diffusion of active monomers does not have any influence on the process of the DLW. When considering diffusion of large molecules like the monomer (PETA, PETTA or TMPTA) or the photoinitiator (molar mass $M \approx 300$ g/mol), values of several hundred picometers are assumed for the hydrodynamic radius (e.g., $R_0 = 750$ pm, which is the radius of a sphere covering the same volume as one PETA molecule). In that case, the diffusion times are even larger, i.e., $\tau_d \approx 20$ ms for PETA. This is longer than typical polymerization duration even in the case of long exposures. Hence,

diffusion of these species is unlikely to result in a significant contribution to the overall process as the reaction duration is too short for diffusion of these molecules to occur. For growing polymer clusters, the values for τ_d become even larger while at the same time, the solvent viscosity increases with the ongoing polymerization. Therefore, diffusion of the relatively large photoinitiator and monomer molecules can be neglected at all later stages of the reaction in most cases. For molecular oxygen, this corresponds to a diffusion time of $\tau_d \approx 2$ ms, so that in the case of longer exposures, oxygen can diffuse into the reaction volume from the surrounding and thereby prevent polymerization. Therefore, it is the molecular oxygen that determines the minimum size of the features that could be produced by the technique (Adv. Optical Mater. 2019, 7, 1901040). The effect of oxygen quenching on resolution and the effect of molecular oxygen on the smaller feature size that can be achieved is the subject of our ongoing study.

Reference: Jonathan B. Müller, 2015, Exploring the Mechanisms of 3D Direct Laser Writing by Multi-Photon Polymerization, Karlsruhe Institute of technology.

Additionally, readers (and Reviewers) would benefit from additional emphasis on limitations of computational resources, that these models and methods are a pioneering framework and that, as compute power grows, the complexity of the MD parts of this model can achieve greater fidelity. These simulations, with $1.2e6$ particles, are big MD simulations compared with many contemporary works, mention of the resources/wall times required for the large-scale simulations would be very interesting.

Reply: As a reference to the computational performance of our CGMD simulation, the printing in a cubic box with $4 \times 4 \times 2$ voxels, and with the minimum exposure of 200τ for a single voxel, takes 20 min on 8 CPUs and 1 GPU. By increasing the exposure time for each voxel to 1200τ the wall time increases to about 120 min. However, for printing more voxels / larger systems, due to the frequent updating in the topology of the polymer network, the scaling performance is reduced: printing of a pyramid with 640 voxels aligned as a 3D pyramid takes more than 2 days. Printing of each rod with 12 voxels takes about 90 minutes.

We have added the above information to the manuscript.

Reviewer 3: The authors express most quantities in reduced Lennard-Jones units ($\epsilon=1$, $\sigma=1$, $m=1$ for the beads), but the power of the laser beam starting the polymerization is given in SI units (from 10 to 50 mW). This mixing of units is confusing and should be avoided. For example, in Figure 2 the exposure time is in τ units (where τ is the LJ time) whereas power is in mW. For the same reason, sentences such as “it is sufficient to apply moderate laser powers of 40-45 mW to reach a high monomer conversion ratio” do not mean much, in the context of this model. The laser power should be given in ϵ/τ units, or alternatively ϵ , σ , m and τ should be in SI units.

Authors' response: Our current model can be considered as a generic model for the tri-acrylate families of resins. Therefore, our aim is not to provide quantitative results on the mechanical properties since there is a significant difference in the corresponding values for different tri-acrylate based resins (but the overall trend is the same (*IEEE Transactions on Nanotechnology*, 16, 23-31, 2017, DOI: 10.1109/TNANO.2016.2625820)). On the other hand, the performance of our model has been evaluated against some experimental data. Therefore, in all cases, the laser power is reported in the real units in order to make it easier for a wider audience to evaluate the model.

Reviewer 4: I agree with the last suggestion of Reviewer 3 and believe that this work will more easily reach experimentalists, as mentioned above, by conversion to SI units. L-J (or other reduced units) will limit the engagement of the interested non-specialist and introduce an unnecessary barrier in a journal with broad readership.

Reply: We thank the reviewer for the comment.

Practically mapping the LJ unit to SI units is rational where the time scale of the coarse-grained simulations is determined by the condition that the coarse-graining and the microscopic model predict similar dynamics on the largest time scales accessible to the microscopic approach. In the present case, the dynamic of the DLW is accelerated similarly to the earlier application of the KG model. This mismatch has been considered as a natural and highly desired consequence of the elimination of microscopic degrees of freedom and of the associated dissipation mechanisms in the KG model and has been discussed in earlier studies (*Macromolecules*, 2020, 53, 1901–1916, *Phys. Mater.* 3, 2020, 034007). In principle, it is possible to match the timescale of the experiment by tuning the friction of a Langevin thermostat such that the resulting τ_K matches the experimental target value. However, this would make the simulations orders of magnitude more expensive in terms of computer time without providing additional physical insight.

Reviewer 3: In connection with the laser power, where does the $\alpha=0.007$ factor come from (in Table 1)?

Authors' response: Most of the parameters on the radiative and polymerization part of the simulations are taken from the numerical model developed to study the polymerization reaction kinetics in direct laser writing by Mueller et al. (*Advanced Materials*. 2014, 26(38), 6566-71 which has been already cited in the manuscript). The numerical model uses both experimental and fitted data. The initiation efficiency ($\alpha=0.007$) is also from the numerical model which reproduces the time-dependent concentrations of the reacting species at the threshold writing laser power. To clarify this also for the readers, we have added more information on this parameter to the manuscript.

Reviewer 4: The authors address Reviewer 3's question adequately.

Reviewer 3: On page 6, it is written: "To study the mechanical properties, we fabricate cubic polymer networks applying different laser powers and constant writing velocities". Again, this is not clear. What are the "cubes" that make up the network?

Authors' response: The cubic polymer networks presented in the first part of the study are representative of one voxel (these cubes are not fabricated voxel by voxel but by uniform irradiation). The reason is that in the first part of the study, our aim is to study the relationship between the laser power (or exposure time) and the degree of monomer conversion. The corresponding data have been used to validate the behavior of our model against the experimental data; therefore, it is reasonable to study the behavior of only one voxel to avoid

the effect of voxel specifications (voxel overlapping, voxel size, shape, etc...) on the degree of monomer conversion.

Reviewer 4: The authors' text and the response was not clear to me either. I assume that by "cubes" the authors used simulation boxes with uniform 'irradiation' and periodic boundary conditions in x, y, and z to simulate a bulk triacrylate thermoset, not a voxel. Is this correct? If so, stating that a "cube" represents a voxel is incorrect since the edge effects of the voxel (the thermoset/liquid interface) are ignored. It is the finite size that defines the voxel. The authors also state elsewhere that investigation of this interface, in the context of writing resolution, will be considered in future work. Therefore, I suggest that this section be revised again for readability.

Reply: We thank the Reviewer for pointing this out. We have clarified it in the manuscript as follows:

"cubic polymer networks of size $40 \times 40 \times 40 \sigma$ with an initial volume fraction of 85% monomers were fabricated by uniform irradiation and applying periodic boundary conditions in all directions."

Reviewer 3: In the "Polymerization model and algorithm" section, the authors do not provide the values of the rate constants k that control the polymerization. The probabilities in Table 1 act as a substitute of the rate constants, but the text does not describe in sufficient detail how these probabilities are employed in the creation of new bonds. Also, what is the value of Θ (number of MD steps between reactions), introduced on page 13?

Authors' response: Thank you very much for pointing this out. We apologize for this inconsistency. The value of $\Theta=10$ was reported as τR in table 1. The τR symbol has been changed to Θ to be unified through the text and table.

Since the method of bond creation by employing the corresponding probabilities is part of the standard "fix bond/create command" implemented in the LAMMPS simulation package, we did not add more explanation on this part to the manuscript as the corresponding information is documented on the LAMMPS official website (https://lammeps.sandia.gov/doc/fix_bond_create.html) under the "fix bond/create" keyword.

Reviewer 4: To permit reproducibility, the use of bond/create should be mentioned explicitly in the main text. The incorporation of the authors' parameters into that fix's implementation are then rather straightforward. The authors should also mention use of SIMSTACK and its relation to LAMMPS in the main text. SIMSTACK appears to be integral to these simulations.

Reply: Thank you very much for your comment.

Due to the limitations of the current version of "fix bond/create" implemented in LAMMPS (for example it does not allow different types of bonds to be created at the same time), we have used a modified version of "fix bond/create" allowing the formation of different types of bonds

according to the reaction scheme illustrated in the manuscript. It has been also used in earlier studies for the simulation of continuous 3D printing (Macromolecules 2017, 50, 19, 7794–7800). We have added the corresponding information to the manuscript as follows in order to clarify the procedure:

“A modified version of “fix bond/create” has been applied in order to allow the formation of multiple types of the bonds at the same time according to the reaction scheme illustrated earlier.”

The use of SIMSTACK, an in-house developed workflow engine, is not compulsory for running the MD simulations of 3D printing. All simulations are reproducible by following manually the procedure described in the methods section. However, SIMSTACK provides an automated way of performing our DLW simulation which is beneficial particularly for non-expert users. The Software is available upon request.

Reviewer 3: At the beginning of the same section, a voxel is described as an “isosurface”. To me, a voxel is a little cubic region, so I do not understand its description as a surface. What is the size of a voxel? How many polymer beads does it contain, on average?

Authors’ response: The definition of the voxel has been revised.

Inspired from the experimental shape of the voxel, in our model, the voxel is not a cubic region but is a prolate spheroidal volume being moved three-dimensionally through the box of free monomers allowing the formation of polymer networks and generating the nano-rods as is also shown in the supplementary movies.

The voxel size is $(48 \times 24 \times 24) \sigma$ and the average number of monomers inside the voxel is about 20000

The above information has been added to the method section.

Reviewer 4: Reviewer 3’s confusion is understandable. Related to my earlier responses, I believe that the manuscript would be strengthened by a clearer description of the two-photon polymerization DLW instrumentation, where the voxel (“volumetric pixel”) has the shape described by the authors, a result of quadratic dependence on laser power. The voxel size and monomer contents questions are answered adequately.

Reply: We thank the reviewer for the suggestion. Following the referee’s suggestions, in the revised manuscript, we have modified Figure 8 in order to provide a better illustration of the DLW process in our model. We have also added the following explanation to the manuscript:

“In the experiment, the nonlinear dependence of two-photon absorption (TPA) on the intensity of light, together with the relatively small TPA cross-section of materials, results in a spatial confinement of the excitation which restricts the polymerization reaction to the focal volume of a high-intensity laser called “voxel”.”

Reviewer 3: The standard Grest-Kremer bead density of 0.85 (in LJ units, see Table 1) is appropriate for systems with purely repulsive bead-bead interactions (WCA potential, truncated at $\sigma=1.1225$). However the authors include the attractive part of the LJ potential (cutoff at $\sigma=2.5$). In my experience, this leads to instabilities, with strong heterogeneities in density (“voids”). A more reasonable value of the density of beads with attractive interaction should be between 0.95 and 1.05.

Authors’ response: Thank you very much for the information. The number density 0.85 was chosen to be compatible with the experimental number density of monomers inside the monomer pool.

Reviewer 4: There may be misunderstanding here. Reviewer 3 is suggesting that the selected number density of 0.85 can lead to instabilities and heterogeneity, not that 0.85 disagrees with experiment. To respond to this, the authors should validate the spatial homogeneity of their simulated resin, particularly considering the authors’ use of the NVE ensemble.

Reply: We thank the reviewer for the clarification. In order to check whether the structure of the cubic polymer networks is homogeneous, we had performed several stress/strain simulations by stretching the cubic polymer networks along different directions (x,y, and z) in order to see whether the direction of stretching has any effect on the stress-strain curve. This is a good test to check the spatial homogeneity of the printed polymer networks. As expected, we have observed small variations between different calculations (an example is shown in the plot for Laser power= 8 mW) indicating the homogeneity of the printed polymer networks .

The following explanation has been added to the manuscript:

“We repeated the procedure and stretched the simulation box along other directions (L_z and L_y) to obtain the mechanical response to uniaxial deformation. Very small variation has been

observed between calculations which indicates that the polymer networks have homogeneous structures.”

Additional Reviewer 4 comment:

Proofreading is required. A few examples:

I believe Figure 3D and E in the main text refer to Figures 3B and C, Figure 2 refers to Figure 1 in one instance. Sentence “No high strain region can be detected for samples with.” requires revision.

Reply: Thank you very much for pointing this out. The mentioned typos have been corrected in the manuscript and the whole manuscript has been checked.

The addition of experimental data in Figure 3C appears to correspond to one of the two experimental data sets in Figure 4 of Reference 49. If so, the authors should explicitly clarify the choice to show one data set over the other or if some other intermediate analysis or data from this reference was used.

Reply: Thank you very much for your comment. In Ref. 49 the authors investigated two photoresist formulations, the difference between them is only the choice of photoinitiator where in formulation A they use B3FL and in formulation B they use bulky R1 as photoinitiators. The experimental data that has been used in our model comes from a photoresist formulation where photoinitiators and monomer molecules have comparable sizes. In reference 49, formulation A meets this condition (in formulation B the photoinitiators are much larger than the monomers). Therefore, we selected experimental data of formulation A which also shows better agreement with our simulation results.

In NCOMMS-21-02461A-Z the authors appear to use the incorrect reaction mechanism. A trifunctional acrylate can form 6 bonds with neighboring monomers, 2 for each acrylate group, not 3 as suggested and implemented in this work. “We find that the increase of laser power leads to a higher cross-linking density (number of the monomers that formed the maximal 3 bonds in this case) as shown in...”

Each acrylate functional group can be within an acrylate chain with 2 neighbors, like in the image below:

Figure 9 / Mechanism of Free Radical Polymerization

Reaction 2, Propagation, in the scheme shows the radical hopping across the molecule from one acrylate site to another (instead of moving to the secondary carbon of the vinyl group), likely the root cause of this error:

This is not the proper mechanism. Even though the entire monomer is represented by a single bead, this bead should have a maximum degree of 6, not 3, as the real monomer exhibits. This difference will result in a significantly different polymer network topology and should be addressed by the authors in the main text.

Reply: We thank the reviewer for pointing out this issue.

We have modified the scheme of radical polymerization in the manuscript as below:

(A)

1. Initiation

2. Propagation

3. Combination
(Termination)

4. Termination
by Oxygen quenching

(B)

However, according to the reaction mechanism provided by the Reviewer (and in general according to the principles of polymer chemistry (Su WF. Radical Chain Polymerization. In: Principles of Polymer Design and Synthesis. Lecture Notes in Chemistry, vol 82. Springer, Berlin, Heidelberg. https://doi.org/10.1007/978-3-642-38730-2_7), each acrylate group is only able to form **1 bond** with the neighboring monomers which means **3 bonds** for a triacrylate monomer (and also 3 bonds for each bead in our simulations).

“Challenges and Limits of Mechanical Stability in 3D Direct Laser Writing” by Elaheh Sedghamiz, Modan Liu, and Wolfgang Wenzel is an exciting contribution, but I believe that this work still suffers from a fundamental issue to be resolved regarding the reaction mechanism, covered at length at the end of this response letter.

We would like to thank the Editor for taking the time to consider our manuscript and the Reviewers for carefully reviewing our manuscript and providing constructive suggestions. On behalf of all authors, I am submitting the revised manuscript titled “Challenges and Limits of Mechanical Stability in 3D Direct Laser Writing” by Elaheh Sedghamiz, Modan Liu and Wolfgang Wenzel in which we address the comments of the reviewers. The point-by-point response to the comments is as follows:

Reviewer #4 (Remarks to the Author):

Reviewer 4 responding in Reviewer 3's absence:

Reviewer 3: Challenges and limits of Mechanical Stability in 3D Direct Laser Writing by Elaheh Sedghamiz, Modan Liu and Wolfgang Wenzel. This manuscript presents molecular dynamics simulation of direct laser writing of polymer networks. This is an interesting and technologically relevant problem. However, the manuscript lacks clarity and I have some doubts about the specifics of the implementation. Below I provide some details. In my opinion, it is not suitable for publication on Nature Communication. At the end of the Introduction, the authors state that they “focus on an experimentally important system”, but they do not let us know what this system is.

Authors' response: By the term “experimentally important system”, we refer to direct laser writing using the tri-acrylate family of photoresists that are being widely used especially for commercial purposes. We have clarified this in the manuscript.

Reviewer 4: This modification was simply the addition of a clause defining the class of monomer studied. The novelty and interesting part of this work is in the simulated irradiation of the thermoset: Rastering the focal region of a laser in 3-dimensions to simulate direct laser writing by two-photon polymerization like in the commercial Nanoscribe instrument. The approach toward simulated polymerization of a trifunctional acrylate is not as novel and has been simulated at the all-atom level in recent work (J. Phys. Chem. B 2020, 124, 9204–9215. <https://doi.org/10.1021/acs.jpcc.0c05319>.) I find this work, simulation of 2PP DLW by coarse-grained MD, more impactful than Reviewers 3 and 2 do, with similar reasoning as Reviewer 1.

Reply: We thank the reviewer for reviewing our manuscript and pointing out the impact of our work and have cited the paper in the revision.

Reviewer 3: They use a generic coarse-grained model to describe the monomers undergoing polymerization. The model is derived from the bead-and-spring polymer model by Grest-Kremer, but it is supplemented with the possibility to form new bonds according to a kinetic scheme. The scheme would benefit from an illustration of the typical molecules and reactions that are being simulated (even if the coarse-grained model cannot reproduce them faithfully). On page 5, it is written: “Fig. 2 shows the kinetics of monomer consumption in one voxel for two different exposure times, i.e. 50 and 500 time steps. For the short exposure time, the polymerization reaction occurs from 0 to 200 τ while for the longer exposure time it occurs between 0 to 170 τ .” This is not very clear to me. Also, in the Grest-Kremer model, the (Langevin)

equations of motion are integrated with a time step of 0.01τ (or lower), but the authors seem to confuse the time step and the LJ time unit.

Authors' response: Thank you very much for pointing this out. We apologize for this typo. All corresponding plot axes have been changed to the LJ time unit. A figure with details about the molecules and reactions has been added.

Reviewer 4: The authors have fully addressed Reviewer 3's concerns.

Reviewer 3: In order to understand the effect of irradiation, which creates active monomers that may diffuse through the medium before undergoing reaction, it would be important to characterize the diffusion properties of these monomers (how far do they travel before they react?). This information seems very important, as it determines the minimum size of the features that could be produced by the technique. However, such information is completely lacking.

Authors' response: Thanks for your comment. In our current approach, the effect of the irradiation results in a local conversion of active monomers, which promptly react with their neighboring monomers if they meet the polymerization criteria specified by the polymerization condition. Since the number density of the monomers in the resin is reasonably high at 0.8, the life-time of active monomers is $\sim 4\tau$ which is extremely short, while the mean free path is measured at $\sim 0.2\sigma$ for the initial stage of the printing, and a reduction to $\sim 0.15\sigma$ as the polymer network grows. In lab conditions, the generation of active monomers only starts after the oxygen in the local vicinity gets depleted, whereas the timescale corresponding to the oxygen diffusion is much smaller than that of the monomers. The investigation of how oxygen diffusion regulates the minimum feature size of printing is the subject of our future study.

The above information has been added to the manuscript.

Reviewer 4: The authors respond qualitatively to Reviewer 3's questions, but these responses highlight a potential shortcoming of this work. The $\sim 0.2\sigma$ should be put into context with estimated travel/mean free path of species in experimental conditions.

Reply: Thank you very much for your comment. In our simulations 0.2σ corresponds to about 0.2 nm and the life time of the active monomers is about 40 ps which is extremely short. In agreement with experimental observations, diffusion of active monomers does not have any influence on the process of the DLW. When considering diffusion of large molecules like the monomer (PETA, PETTA or TMPTA) or the photoinitiator (molar mass $M \approx 300$ g/mol), values of several hundred picometers are assumed for the hydrodynamic radius (e.g., $R_0 = 750$ pm, which is the radius of a sphere covering the same volume as one PETA molecule). In that case, the diffusion times are even larger, i.e., $\tau_d \approx 20$ ms for PETA. This is longer than typical polymerization duration even in the case of long exposures. Hence, diffusion of these species is unlikely to result in a significant contribution to the overall process as the reaction duration is too short for diffusion of these molecules to occur. For growing polymer clusters, the values for τ_d become even larger while at the same time, the solvent viscosity increases with the ongoing polymerization. Therefore, diffusion of the relatively large photoinitiator and monomer molecules can be neglected at all later

stages of the reaction in most cases. For molecular oxygen, this corresponds to a diffusion time of $\tau_d \approx 2$ ms, so that in the case of longer exposures, oxygen can diffuse into the reaction volume from the surrounding and thereby prevent polymerization. Therefore, it is the molecular oxygen that determines the minimum size of the features that could be produced by the technique (Adv. Optical Mater. 2019, 7, 1901040). The effect of oxygen quenching on resolution and the effect of molecular oxygen on the smaller feature size that can be achieved is the subject of our ongoing study. Reference: Jonathan B. Müller, 2015, Exploring the Mechanisms of 3D Direct Laser Writing by Multi-Photon Polymerization, Karlsruhe Institute of technology.

Additionally, readers (and Reviewers) would benefit from additional emphasis on limitations of computational resources, that these models and methods are a pioneering framework and that, as compute power grows, the complexity of the MD parts of this model can achieve greater fidelity. These simulations, with $1.2e6$ particles, are big MD simulations compared with many contemporary works, mention of the resources/wall times required for the large-scale simulations would be very interesting.

Reply: As a reference to the computational performance of our CGMD simulation, the printing in a cubic box with $4 \times 4 \times 2$ voxels, and with the minimum exposure of 200τ for a single voxel, takes 20 min on 8 CPUs and 1 GPU. By increasing the exposure time for each voxel to 1200τ the wall time increases to about 120 min. However, for printing more voxels / larger systems, due to the frequent updating in the topology of the polymer network, the scaling performance is reduced: printing of a pyramid with 640 voxels aligned as a 3D pyramid takes more than 2 days. Printing of each rod with 12 voxels takes about 90 minutes.

We have added the above information to the manuscript.

The authors have adequately addressed this point.

Reviewer 3: The authors express most quantities in reduced Lennard-Jones units ($\epsilon=1$, $\sigma=1$, $m=1$ for the beads), but the power of the laser beam starting the polymerization is given in SI units (from 10 to 50 mW). This mixing of units is confusing and should be avoided. For example, in Figure 2 the exposure time is in τ units (where τ is the LJ time) whereas power is in mW. For the same reason, sentences such as “it is sufficient to apply moderate laser powers of 40-45 mW to reach a high monomer conversion ratio” do not mean much, in the context of this model. The laser power should be given in ϵ/τ units, or alternatively ϵ , σ , m and τ should be in SI units.

Authors' response: Our current model can be considered as a generic model for the tri-acrylate families of resins. Therefore, our aim is not to provide quantitative results on the mechanical properties since there is a significant difference in the corresponding values for different tri-acrylate based resins (but the overall trend is the same (*IEEE Transactions on Nanotechnology*, 16, 23-31, 2017, DOI: 10.1109/TNANO.2016.2625820)). On the other hand, the performance of our model has been evaluated against some experimental data. Therefore, in all cases, the laser power is reported in the real units in order to make it easier for a wider audience to evaluate the model.

Reviewer 4: I agree with the last suggestion of Reviewer 3 and believe that this work will more easily reach experimentalists, as mentioned above, by conversion to SI units. L-J (or other reduced units) will limit the engagement of the interested non-specialist and introduce an unnecessary barrier in a journal with broad readership.

Reply: We thank the reviewer for the comment.

Practically mapping the LJ unit to SI units is rational where the time scale of the coarse-grained simulations is determined by the condition that the coarse-graining and the microscopic model predict similar dynamics on the largest time scales accessible to the microscopic approach. In the present case, the dynamic of the DLW is accelerated similarly to the earlier application of the KG model. This mismatch has been considered as a natural and highly desired consequence of the elimination of microscopic degrees of freedom and of the associated dissipation mechanisms in the KG model and has been discussed in earlier studies (Macromolecules, 2020, 53, 1901–1916, Phys. Mater. 3, 2020, 034007). In principle, it is possible to match the timescale of the experiment by tuning the friction of a Langevin thermostat such that the resulting τ_K matches the experimental target value. However, this would make the simulations orders of magnitude more expensive in terms of computer time without providing additional physical insight.

A very clear explanation, distillation of this information into a few sentences and adding to the manuscript would be very valuable.

Reviewer 3: In connection with the laser power, where does the $\alpha=0.007$ factor come from (in Table 1)?

Authors' response: Most of the parameters on the radiative and polymerization part of the simulations are taken from the numerical model developed to study the polymerization reaction kinetics in direct laser writing by Mueller et al. (Advanced Materials. 2014, 26(38), 6566-71 which has been already cited in the manuscript). The numerical model uses both experimental and fitted data. The initiation efficiency ($\alpha=0.007$) is also from the numerical model which reproduces the time-dependent concentrations of the reacting species at the threshold writing laser power. To clarify this also for the readers, we have added more information on this parameter to the manuscript.

Reviewer 4: The authors address Reviewer 3's question adequately.

Reviewer 3: On page 6, it is written: "To study the mechanical properties, we fabricate cubic polymer networks applying different laser powers and constant writing velocities". Again, this is not clear. What are the "cubes" that make up the network?

Authors' response: The cubic polymer networks presented in the first part of the study are representative of one voxel (these cubes are not fabricated voxel by voxel but by uniform irradiation). The reason is that in the first part of the study, our aim is to study the relationship between the laser power (or exposure time) and the degree of monomer conversion. The corresponding data have been used to validate the behavior of our model against the experimental data; therefore, it is reasonable to study the behavior of only one voxel to avoid the effect of voxel specifications (voxel overlapping, voxel size, shape, etc...) on the degree of monomer conversion.

Reviewer 4: The authors' text and the response was not clear to me either. I assume that by "cubes" the authors used simulation boxes with uniform 'irradiation' and periodic boundary conditions in x, y, and z to simulate a bulk triacrylate thermoset, not a voxel. Is this correct? If so, stating that a "cube" represents a voxel is incorrect since the edge effects of the voxel (the thermoset/liquid interface) are ignored. It is the finite size that defines the voxel. The authors also state elsewhere that investigation of this interface, in the context of writing resolution, will be considered in future work. Therefore, I suggest that this section be revised again for readability.

Reply: We thank the Reviewer for pointing this out. We have clarified it in the manuscript as follows:

"cubic polymer networks of size $40 \times 40 \times 40 \sigma$ with an initial volume fraction of 85% monomers were fabricated by uniform irradiation and applying periodic boundary conditions in all directions."

The authors have adequately addressed this point.

Reviewer 3: In the "Polymerization model and algorithm" section, the authors do not provide the values of the rate constants k that control the polymerization. The probabilities in Table 1 act as a substitute of the rate constants, but the text does not describe in sufficient detail how these probabilities are employed in the creation of new bonds. Also, what is the value of Θ (number of MD steps between reactions), introduced on page 13?

Authors' response: Thank you very much for pointing this out. We apologize for this inconsistency. The value of $\Theta=10$ was reported as τ_R in table 1. The τ_R symbol has been changed to Θ to be unified through the text and table.

Since the method of bond creation by employing the corresponding probabilities is part of the standard "fix bond/create command" implemented in the LAMMPS simulation package, we did not add more explanation on this part to the manuscript as the corresponding information is documented on the LAMMPS official website (https://lammps.sandia.gov/doc/fix_bond_create.html) under the "fix bond/create" keyword.

Reviewer 4: To permit reproducibility, the use of bond/create should be mentioned explicitly in the main text. The incorporation of the authors' parameters into that fix's implementation are then rather straightforward. The authors should also mention use of SIMSTACK and its relation to LAMMPS in the main text. SIMSTACK appears to be integral to these simulations.

Reply: Thank you very much for your comment.

Due to the limitations of the current version of "fix bond/create" implemented in LAMMPS (for example it does not allow different types of bonds to be created at the same time), we have used a modified version of "fix bond/create" allowing the formation of different types of bonds according to the reaction scheme illustrated in the manuscript. It has been also used in earlier studies for the simulation of continuous 3D printing (Macromolecules 2017, 50, 19, 7794–7800). We have added the corresponding information to the manuscript as follows in order to clarify the procedure:

“A modified version of “fix bond/create” has been applied in order to allow the formation of multiple types of the bonds at the same time according to the reaction scheme illustrated earlier.”

The use of SIMSTACK, an in-house developed workflow engine, is not compulsory for running the MD simulations of 3D printing. All simulations are reproducible by following manually the procedure described in the methods section. However, SIMSTACK provides an automated way of performing our DLW simulation which is beneficial particularly for non-expert users. The Software is available upon request.

Thanks! The authors have adequately addressed this point.

Reviewer 3: At the beginning of the same section, a voxel is described as an “isosurface”. To me, a voxel is a little cubic region, so I do not understand its description as a surface. What is the size of a voxel? How many polymer beads does it contain, on average?

Authors’ response: The definition of the voxel has been revised. Inspired from the experimental shape of the voxel, in our model, the voxel is not a cubic region but is a prolate spheroidal volume being moved three-dimensionally through the box of free monomers allowing the formation of polymer networks and generating the nano-rods as is also shown in the supplementary movies.

The voxel size is $(48 \times 24 \times 24) \sigma$ and the average number of monomers inside the voxel is about 20000

The above information has been added to the method section.

Reviewer 4: Reviewer 3’s confusion is understandable. Related to my earlier responses, I believe that the manuscript would be strengthened by a clearer description of the two-photon polymerization DLW instrumentation, where the voxel (“volumetric pixel”) has the shape described by the authors, a result of quadratic dependence on laser power. The voxel size and monomer contents questions are answered adequately.

Reply: We thank the reviewer for the suggestion. Following the referee’s suggestions, in the revised manuscript, we have modified Figure 8 in order to provide a better illustration of the DLW process in our model. We have also added the following explanation to the manuscript:

“In the experiment, the nonlinear dependence of two-photon absorption (TPA) on the intensity of light, together with the relatively small TPA cross-section of materials, results in a spatial confinement of the excitation which restricts the polymerization reaction to the focal volume of a high-intensity laser called “voxel”.”

The authors have adequately addressed this point.

Reviewer 3: The standard Grest-Kremer bead density of 0.85 (in LJ units, see Table 1) is appropriate for systems with purely repulsive bead-bead interactions (WCA potential, truncated at $\sigma=1.1225$). However the authors include the attractive part of the LJ potential (cutoff at $\sigma=2.5$). In my experience, this leads to instabilities, with

strong heterogeneities in density (“voids”). A more reasonable value of the density of beads with attractive interaction should be between 0.95 and 1.05.

Authors’ response: Thank you very much for the information. The number density 0.85 was chosen to be compatible with the experimental number density of monomers inside the monomer pool.

Reviewer 4: There may be misunderstanding here. Reviewer 3 is suggesting that the selected number density of 0.85 can lead to instabilities and heterogeneity, not that 0.85 disagrees with experiment. To respond to this, the authors should validate the spatial homogeneity of their simulated resin, particularly considering the authors’ use of the NVE ensemble.

Reply: We thank the reviewer for the clarification. In order to check whether the structure of the cubic polymer networks is homogeneous, we had performed several stress/strain simulations by stretching the cubic polymer networks along different directions (x,y, and z) in order to see whether the direction of stretching has any effect on the stress-strain curve. This is a good test to check the spatial homogeneity of the printed polymer networks. As expected, we have observed small variations between different calculations (an example is shown in the plot for Laser power= 8 mW) indicating the homogeneity of the printed polymer networks .

The following explanation has been added to the manuscript:

“We repeated the procedure and stretched the simulation box along other directions (L_z and L_y) to obtain the mechanical response to uniaxial deformation. Very small variation has been observed between calculations which indicates that the polymer networks have homogeneous structures.”

The authors have adequately addressed this point.

Additional Reviewer 4 comment:

Proofreading is required. A few examples:

I believe Figure 3D and E in the main text refer to Figures 3B and C, Figure 2 refers to Figure 1 in one instance. Sentence “No high strain region can be detected for samples with.” requires revision.

Reply: Thank you very much for pointing this out. The mentioned typos have been corrected in the manuscript and the whole manuscript has been checked.

The authors have adequately addressed this point.

The addition of experimental data in Figure 3C appears to correspond to one of the two experimental data sets in Figure 4 of Reference 49. If so, the authors should explicitly clarify the choice to show one data set over the other or if some other intermediate analysis or data from this reference was used.

Reply: Thank you very much for your comment. In Ref. 49 the authors investigated two photoresist formulations, the difference between them is only the choice of photoinitiator where in formulation A they use B3FL and in formulation B they use bulky R1 as photoinitiators. The experimental data that has been used in our model comes from a photoresist formulation where photoinitiators and monomer molecules have comparable sizes. In reference 49, formulation A meets this condition (in formulation B the photoinitiators are much larger than the monomers). Therefore, we selected experimental data of formulation A which also shows better agreement with our simulation results.

Again, a very clear explanation and noting this in the manuscript would be of great value to the interested reader.

In NCOMMS-21-02461A-Z the authors appear to use the incorrect reaction mechanism. A trifunctional acrylate can form 6 bonds with neighboring monomers, 2 for each acrylate group, not 3 as suggested and implemented in this work. “We find that the increase of laser power leads to a higher cross-linking density (number of the monomers that formed the maximal 3 bonds in this case) as shown in...”

Each acrylate functional group can be within an acrylate chain with 2 neighbors, like in the image below:

Figure 9 / Mechanism of Free Radical Polymerization

Reaction 2, Propagation, in the scheme shows the radical hopping across the molecule from one acrylate site to another (instead of moving to the secondary carbon of the vinyl group), likely the root cause of this error:

This is not the proper mechanism. Even though the entire monomer is represented by a single bead, this bead should have a maximum degree of 6, not 3, as the real monomer exhibits. This difference will result in a significantly different polymer network topology and should be addressed by the authors in the main text.

Reply: We thank the reviewer for pointing out this issue.

We have modified the scheme of radical polymerization in the manuscript as below:

(A)

1. Initiation

2. Propagation

3. Combination
(Termination)

4. Termination
by Oxygen quenching

(B)

However, according to the reaction mechanism provided by the Reviewer (and in general according to the principles of polymer chemistry (Su WF. Radical Chain Polymerization. In: Principles of Polymer Design and Synthesis. Lecture Notes in Chemistry, vol 82. Springer, Berlin, Heidelberg. https://doi.org/10.1007/978-3-642-38730-2_7), each acrylate group is only able to form **1 bond** with the neighboring monomers which means **3 bonds** for a triacrylate monomer (and also 3 bonds for each bead in our simulations).

I believe the authors are incorrect and will provide a more thorough explanation. Consider a simple monofunctional acetate, benzyl acetate:

Figure A.

Benzyl acetate can polymerize to poly(benzyl acrylate):

Figure B.

The monoacrylate in Figure B forms bonds with 2 neighboring acrylate monomers, indicated by the two asterisks on either side of the square brackets. The image above represents a linear chain, similar to the more classic example of polystyrene the polymer of styrene (vinyl benzene).

A general poly [mono]acrylate is:

Figure C.

Two inter-monomer bonds per acrylate. Linear chain. The two bonds are on either side of the square brackets.

Addition of a second acrylate group allows for the formation of heavily crosslinked polymers, e.g. HDDA, shown in Figure D, is basically a hexane with an acrylate group on each end. HDDA is a crosslinker that can participate in two linear chains, each end similar to Figures A-C: 2 bonds per acrylate group. 4 total inter-monomer bonds, connected to R1-R4.

Figure D.

The authors assert that “each acrylate group is only able to form **1 bond** with the neighboring monomers which means **3 bonds** for a triacrylate monomer.” A trifunctional acrylate, like the one simulated in this work, has three pendant acrylate groups.

Acrylate polymerization is hopefully more clear after showing the simpler examples in Figures A-C. Multiple functionality is shown in Figure D. The monofunctional & difunctional acrylate examples can be expanded to the trifunctional monomer simulated in this work:

Figure E.

Where each pendant acrylate group is participating in an acrylate chain, 2 acrylate neighbors on each of the 3 pendant acrylate groups. Positions of neighbors indicated by R1-R6. A coarse-grained ‘bead’ of this molecule must therefore have a maximum of 6 inter-monomer linkages to properly simulate microscopic topology.